# Revisiting Confidence Calibration for Misclassification Detection in VLMs

**Jincheng Huang**[1]   **Jie Xu**[4]   **Xiaoshuang Shi**[1]   **Ping Hu**[1]   **Lei Feng**[2*]   **Xiaofeng Zhu**[3*]

[1]School of Computer Science and Engineering,
University of Electronic Science and Technology of China
[2]Southeast University, [3]Hainan University, [4]Singapore University of Technology and Design

## Abstract

Confidence calibration has been widely studied to improve the trustworthiness of predictions in vision-language models (VLMs). However, we theoretically reveal that standard confidence calibration inherently *impairs* the ability to distinguish between correct and incorrect predictions (*i.e.,* Misclassification Detection, MisD), which is crucial for reliable deployment of VLMs in high-risk applications. In this paper, we investigate MisD in VLMs and propose confidence recalibration to enhance MisD. Specifically, we design a new confidence calibration objective to replace the standard one. This modification theoretically achieves higher precision in the MisD task and reduces the mixing of correct and incorrect predictions at every confidence level, thereby overcoming the limitations of standard calibration for MisD. As the calibration objective is not differentiable, we introduce a differentiable surrogate loss to enable better optimization. Moreover, to preserve the predictions and zero-shot ability of the original VLM, we develop a post-hoc framework, which employs a lightweight meta network to predict sample-specific temperature factors, trained with the surrogate loss. Extensive experiments across multiple metrics validate the effectiveness of our approach on MisD.

## 1 Introduction

Pretrained Vision-language models (VLMs) (Radford et al., 2021; Zhou et al., 2022b;a; Khattak et al., 2023), such as CLIP (Radford et al., 2021), have demonstrated impressive zero-shot capabilities. Owing to their strong generalization ability and pretrained nature, they have been applied to a wide range of downstream tasks, including autonomous driving (Cui et al., 2024), medical diagnosis (Zhao et al., 2023), and 3D scene understanding (Chen et al., 2023). While these models improve flexibility and accuracy, ensuring their reliability remains essential, which is crucial for real-world deployment and safety-critical applications. Consequently, confidence calibration (Guo et al., 2017), which adjusts model confidence to better match true correctness, is therefore an important component in developing reliable VLM-based systems.

Confidence calibration aims to align the model's predicted confidence with the true likelihood of correctness. A classical calibration method is Temperature Scaling (Guo et al., 2017), which adjusts the sharpness of the output probabilities using a temperature coefficient to better align confidence with empirical accuracy. However, a single global temperature overlooks the instance-wise variation in confidence miscalibration. Therefore, many instance-wise calibration methods have been proposed (Huang et al., 2025; Krishnan & Tickoo, 2020). In VLMs, confidence calibration becomes more challenging due to modality-specific factors or the process of fine-tuning. To address these issues, several works (Wang et al., 2024; Lv et al., 2025) show that distances in the text embedding space play a crucial role in calibration errors, and they incorporate this text-modality signal as an additional temperature adjustment. In contrast, to mitigate miscalibration introduced by fine-tuning, DOR (Wang et al., 2025) uses a large vocabulary set to preserve the semantic structure of the pretrained CLIP, thereby reducing the shift in text features caused by prompt tuning.

Although previous methods can achieve great calibration for VLMs, many high-risk tasks rely more on the model's ability to rank predictions by correctness than confidence calibration performance.

---

*Corresponding authors

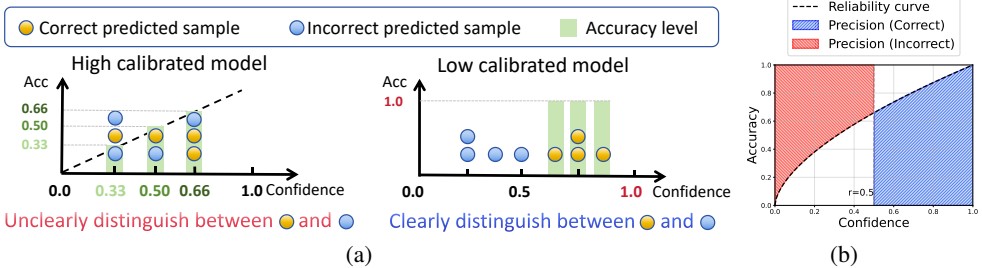

Figure 1: (a) Comparison between a calibrated model and one with improved MisD performance. Left: a calibrated model aligns predicted confidence with accuracy. Right: a model with higher MisD performance that better ranks correct predictions above incorrect ones. (b) Illustration of the relationship between the reliability curve and the MisD. Given a reliability curve and a confidence threshold $r$, samples with confidence above $r$ are regarded as correct predictions. Their precision can be derived from the area under the curve within that interval. Conversely, samples with confidence below $r$ are treated as incorrect predictions, and their precision can be derived from the area above the curve.

Concretely, in the misclassification detection task, it is more desirable for the VLMs to consistently assign lower confidence scores to misclassified samples than to correctly classified ones, rather than merely aligning predicted probabilities with accuracy, which is illustrated in Figure 1(a).

In this work, we first analyze the reliability diagram of confidence calibration for MisD, and find that the region under (above) the reliability curve is related to the precision of detecting correct (incorrect) prediction, as shown in Figure 1(b). Motivated by this insight, we reveal that standard calibration inherently limits the upper bound of this precision by analyzing its reliability curve. To remedy this limitation, we introduce a new target curve. It increases the area under the curve in the high-confidence region and enlarges the area above the curve in the low-confidence region, as illustrated in Figure 2. This design theoretically improves precision and reduces the mixing between correctly and incorrectly classified samples, thereby encouraging better separation between them. Although the proposed reliability curve is ideal for MisD, it cannot be directly expressed as a differentiable loss function. To bridge this gap, we design a surrogate loss that faithfully captures its effect, making the objective practically optimizable. In practice, to avoid interfering with the underlying VLM predictions, we adopt a post-hoc framework. Specifically, we introduce a lightweight meta network that predicts the temperature scaling factor to each sample, conditioned on the VLM logits, image embeddings, and predicted text embeddings. This meta network is trained using our surrogate loss. Compared to previous works, our contributions are listed as follows:

- We make the first attempt to address MisD in VLMs from a calibration perspective. By analyzing the reliability diagram for MisD, we uncover key limitations of standard calibration and introduce a new perspective to systematically analyze MisD and guide its improvement.

- We propose a new calibration objective tailored to improving MisD performance. The theoretical analysis guarantees that the proposed objective can improve the precision in the MisD task and reduce mixing of correct and incorrect predictions on every confidence level.

- To ensure that the proposed loss does not interfere with the pretrained capabilities of VLMs, we develop a post-hoc calibration framework that learns individualized temperature coefficients for each sample. Empirical results show that the proposed method consistently improves MisD over the recent uncertainty estimation methods across diverse settings.

## 2  RELATED WORK

This section briefly reviews the topics related to this work, including confidence calibration and misclassification detection, and the recent works in VLMs.

## 2.1 Confidence Calibration

Trustworthy machine learning (Yang et al., 2025c;b; Fang et al., 2025; Zhuo et al., 2025; Yang et al., 2025a; Fang et al., 2026) encompasses various approaches, among which confidence calibration is a key component for ensuring reliable predictive confidence. The calibration methods can be divided into two categories: training-time calibration and post-hoc calibration. For the training-time calibration, a notable example is focal loss (Mukhoti et al., 2020), with subsequent works such as adaptive focal loss (Ghosh et al., 2022) modifying hyperparameters for different sample groups based on prior training knowledge. For the post-hoc methods, the most commonly used method is temperature scaling (TS) (Guo et al., 2017), which proposes to use a hyperparameter (*i.e.,* temperature coefficient) as the denominator of logits, making the confidence adjustable. However, TS is not flexible enough, since TS uses a unified temperature coefficient for every sample. Therefore, many subsequent methods (Wang et al., 2021; Xiong et al., 2023; Yang et al., 2023a;b) aim to improve TS by applying adaptive temperature parameters, treating samples differently for a more effective maximum-entropy regularizer.

In VLMs, calibration becomes more challenging due to modality-specific factors or the process of fine-tuning (Wang et al., 2024; 2025). Consequently, DAC (Wang et al., 2024) utilizes the proximity of the text modality to adjust confidence accordingly. DOR (Wang et al., 2025) mitigating miscalibration introduced during fine-tuning by introducing an extra-large vocabulary set. Although many calibration methods are proposed, the fundamental goal of calibration is to enable a model to distinguish between correct predictions and incorrect predictions. However, in this work, we reveal that even a perfectly calibrated model remains fundamentally limited, and thus may still fail to identify the misclassified samples or correctly predicted samples that bring the potential risk in many scenarios.

## 2.2 Misclassification Detection

Misclassification detection (Hendrycks & Gimpel, 2016), which is also called failure prediction, and the goal is to detect incorrect predictions from correct predictions. It is crucial for machine learning models deployed in high-risk scenarios. Calibration methods (Zhang et al., 2023) are often viewed as an effective strategy for misclassification detection, as aligning predicted confidence with the true likelihood of correctness allows the model to distinguish between reliable and unreliable predictions. However, empirical studies (Zhu et al., 2022) have shown that commonly used calibration methods only bring limited improvement for misclassification detection. Yet, no theoretical analysis has been conducted to uncover the underlying reasons, and consequently, the proposed method of this work has not been optimized from a calibration perspective. Therefore, to the best of our knowledge, no prior work has systematically investigated misclassification detection from the calibration perspective. Instead, existing approaches typically focus on confidence regression (Corbière et al., 2019), exposing outlier samples (Zhu et al., 2023), flat minima (Zhu et al., 2022), among other techniques.

In VLMs, a few recent efforts have been devoted to misclassification detection. For instance, FSMisD (Zeng et al., 2025) adopts a prompt-based strategy, which, however, overlooks the use of confidence information, limiting flexibility and making it difficult to integrate with existing prompt-tuning models. ViLU (Lafon et al., 2025), on the other hand, formulates uncertainty modeling as a binary classification problem, which essentially ignores the role of confidence in distinguishing low-confidence correct predictions from high-confidence misclassifications, and lacks theoretical guarantees. While these approaches provide useful heuristics, they mainly rely on task-specific designs or empirical observations, resulting in limited ability to capture the fundamental role of confidence in misclassification detection.

## 3 Preliminaries

**Contrastive Language-Image Pretraining (CLIP).** CLIP is a powerful vision-language model that aligns image and text representations in a shared embedding space through contrastive learning (Radford et al., 2021). Owing to its contrastive training on large-scale image-text pairs, CLIP demonstrates strong zero-shot generalization and is readily deployable in various downstream scenarios. Let $\xi : \mathbf{x} \to \mathbb{R}^d$ and $\psi : \mathbf{t} \to \mathbb{R}^d$ denote the image and text encoders of CLIP, respectively. Given an image instance $v$ and a text label $c$, the output logit of CLIP can be formulated as:

$$z_{v,c} = \tau_{\text{clip}} \cdot \text{sim}(\xi(\mathbf{x}_v), \psi(\mathbf{t}_c)), \tag{1}$$

where $\mathrm{sim}(\cdot, \cdot)$ denotes cosine similarity, $\mathbf{t}_c$ is a hand-crafted prompt, typically set to "a photo of a {class}" and $\tau_{\mathrm{clip}}$ is a fixed constant, usually set to 100. In multi-class classification, let $C = \{c_0, c_1, \ldots, c_{|C|-1}\}$ denote the set of candidate classes. The predicted label corresponds to the class with the highest predicted probability: which can be formally expressed as follows:

$$\hat{y} = \underset{c \in \mathcal{C}}{\arg\max}(e^{z_c} / \sum\nolimits_{i=0}^{|C|-1} e^{z_i}), \tag{2}$$

where $\hat{y}$ is the predicted class and the associated probability (*i.e.*, $s = \max(e^{z_c} / \sum_{i=0}^{|C|-1} e^{z_i})$) is referred to as the confidence of the prediction.

**Confidence Calibration.** Confidence calibration is particularly important in high-risk and open-world scenarios (Guo et al., 2017). It refers to alignment between a model's predicted confidence and its actual accuracy, making the predicted confidence more trustworthy. For example, if the average confidence is 0.8, then approximately 80% of the predicted examples should be correct. Formally, the objective of confidence calibration can be defined as follows:

$$\mathbb{P}(\hat{y} = y | s = p) = p, \forall p \in [0, 1], \tag{3}$$

where $y$ is the ground-truth label. The performance of confidence calibration can be evaluated using a **reliability diagram**, where the vertical axis represents accuracy and the horizontal axis represents confidence, as shown in Figure 1(b). Points closer to the diagonal indicate better calibration. The **reliability curve** is obtained by smoothly connecting these points, with the diagonal line representing the perfect-calibration curve (*i.e.*, $f(x) = x$). Confidence calibration aims to make the model's reliability curve as close as possible to the perfect calibration curve.

**Misclassification detection (MisD).** Misclassification detection is a critical safeguard for deploying models in real-world applications (Hendrycks & Gimpel, 2016), aiming to distinguish incorrect predictions from correct predictions based on confidence ranking. Formally, given a confidence threshold $r \in [0, 1]$, predictions with confidence above $r$ are detected as correct predictions, while those with confidence below $r$ are detected as incorrect predictions.

## 4 METHOD

**Overview.** The overall research framework can be divided into three parts: ❶ We first analyze the upper bound of calibration for MisD, demonstrating its inherent limitation. To overcome this limitation, we redesign the calibration reliability curve tailored for MisD. In addition, we establish several favorable properties of the proposed curve, highlighting its theoretical soundness and practical advantages in improving MisD. ❷ As the reliability curve is not differentiable, we design a surrogate loss (*i.e.*, $\mathcal{L}_{\mathrm{SUR}}$) to realize its optimization. ❸ Finally, we present a post-hoc and lightweight implementation, which adjusts confidence without modifying the VLMs' parameters.

### 4.1 REVISITING CALIBRATION WITH MISD

The reliability diagram is a key tool for evaluating the calibration of probabilistic classifiers, where the diagonal ($\mathrm{acc}(s) = s$) denotes perfect calibration. Thus, calibration quality is assessed by how closely the reliability curve aligns with this line. So far, it has been used exclusively for this purpose. In this work, we find that the reliability diagram also encodes information relevant to MisD, in particular that the **regions under and above the reliability curve relate to the precision of detecting correct and incorrect predictions.** This insight reveals a direct link between the reliability diagram and MisD.

Specifically, we first analyze the meaning of the region under the reliability curve. Since the vertical axis of reliability diagram denotes accuracy, each curve value reflects the proportion of correct predictions among all samples at that confidence. The enclosed region naturally accumulates these proportions across the confidence spectrum. Building on this view, we formalize the following relationship. The formal proof is listed in Appendix A.1:

**Lemma 4.1.** *Given a confidence interval $[a, b]$, let $w(s)$ denote the density of the sample at confidence level $s$. Then, the precision of correct predictions within $[a, b]$ can be derived from region under the reliability curve $f(s)$ over $[a, b]$,* i.e., $\mathrm{Prec}^+ = (\int_a^b w(s)f(s)ds) / \int_a^b w(s)ds$. *Similarly, the precision of incorrect predictions over $[a, b]$ can be derived from the region above the curve via* $\mathrm{Prec}^- = (\int_a^b w(s)(1 - f(s))ds) / \int_a^b w(s)ds$.

Lemma 4.1 establishes a strong connection between the region under (above) the reliability curve and the task of detecting correct (incorrect) predictions. Since MisD can be quantified by the joint precision of detecting high-confidence predictions (*i.e.*, $s \geq r$) as correct and low-confidence predictions (*i.e.*, $s < r$) as incorrect, it can be characterized by the region below the reliability curve in the high-confidence interval and the region above it in the low-confidence interval.

Based on the above observations, we can examine the effectiveness of confidence calibration for MisD. Specifically, we analyze the precision of the perfect calibration curve in detecting correct and incorrect predictions, as formalized in the following theorem (the proof is listed in Appendix A.3):

**Theorem 4.2.** *Let* $r \in [0, 1]$ *be a confidence threshold. Under perfect calibration, the precision for the correct prediction detection and incorrect prediction detection tasks is* $\mathrm{Prec}^+ = \mathbb{E}_{s \sim w(s | s \in [r, 1])}[s]$ *and* $\mathrm{Prec}^- = \mathbb{E}_{s \sim w(s | s \in [0, r])}[1 - s]$, *respectively.*

Theorem 4.2 demonstrates that, even under the goal of confidence calibration (*i.e.*, perfect calibration), the precision of correct prediction detection equals the conditional expectation of the confidence $s$ in $[r, 1]$. In practice, this expectation is strictly less than 1 **unless all samples are concentrated at confidence** 1, which rarely occurs. Similarly, the precision of incorrect prediction detection corresponds to the conditional expectation of $1 - s$ in $[0, r]$, and is strictly less than 1 **unless all samples are concentrated at confidence 0**. In realistic scenarios, such as with pretrained CLIP models, test samples are often drawn from diverse distributions, causing confidence values to be widely spread across $[0, 1]$ (see Appendix D.8 for empirical validation). Consequently, both the precision of correct and incorrect prediction detection are far below 1. This observation explains why strict calibration alone is insufficient to achieve high MisD performance.

Motivated by Lemma 4.1 and Theorem 4.2, we seek a reliability curve as the calibration objective that explicitly guides optimization toward better MisD. Let $f : [0, 1] \rightarrow [0, 1]$ denote the reliability curve. Based on the above discussion, we therefore formulate the following MisD-oriented objective:

$$\max_{f \in \mathcal{F}} \left( \int_0^{0.5} w(s)[1 - f(x)]dx + \int_{0.5}^1 w(s)f(x)dx \right), \tag{4}$$

where $\mathcal{F} = \{ f : [0, 1] \rightarrow [0, 1], f(0) = 0, f(1) = 1, f \text{ nondecreasing} \}$ is a function family. This objective maximizes the region above the curve in the low-confidence region $[0, 0.5]$, and the region under the curve in the high-confidence region $[0.5, 1]$, thereby improving the precision of correct and incorrect prediction detection. However, directly maximizing Eq.(4) over $\mathcal{F}$ causes overly aggressive probability adjustments and instability. To balance improvement in MisD with smoothness, we impose an additional requirement that $f$ should have a controllable and gradual transition. As a practical and analytically tractable instantiation, we adopt the normalized sigmoid curve:

$$\Psi(s) \triangleq \frac{\sigma\left(\frac{s - 0.5}{\lambda}\right) - \sigma\left(\frac{-0.5}{\lambda}\right)}{\sigma\left(\frac{0.5}{\lambda}\right) - \sigma\left(\frac{-0.5}{\lambda}\right)}, \quad \sigma(z) \triangleq \frac{1}{1 + e^{-z}}, \tag{5}$$

where $\lambda \in \mathbb{R}^+$ controls the smoothness of the transition, enabling a trade-off between separation strength and stability. Figure 2 illustrates the curve for different values of $\lambda$, highlighting its flexibility. We summarize the key properties of the proposed curve that are effective for MisD as follows.

**Property 1a: Perfect calibration as a special case ($\lambda \rightarrow \infty$).** As $\lambda \rightarrow \infty$, $\Psi(x)$ converges uniformly to the diagonal $\Psi(x) = x$ on $[0, 1]$, which corresponds to perfect calibration.

**Property 1b: Step-function limit that maximizes MisD effectiveness ($\lambda \rightarrow 0$).** As $\lambda \rightarrow 0$, $\Psi(x)$ converges to the closed-form solution of Eq.(4) (*i.e.*, step function), then the following corollary holds. The proof is listed in Appendix A.2.

**Corollary 4.3.** *When* $\lambda \rightarrow 0$, *the proposed normalized sigmoid curve assigns higher confidence to all correctly classified samples than to any misclassified sample, i.e.,*

$$\mathbb{P}\big(\mathrm{Conf}(N^+) > \mathrm{Conf}(N^-)\big) = 1, \tag{6}$$

*where* $N^+$ *and* $N^-$ *denote the sets of correctly classified and misclassified samples, respectively. In this case, the reliability curve approaches a step-like curve (cf. Figure 2 with* $\lambda = 1e - 5$).

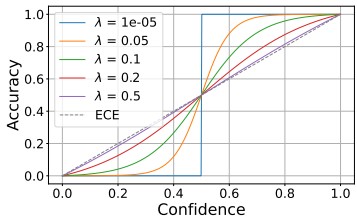

Figure 2: Visualization of the proposed normalized sigmoid curve with different $\lambda$ values alongside ECE in the reliability diagram.

**Properties 1a** and **1b** characterize the effect of the limit behavior as $\lambda$ varies. A smaller $\lambda$ leads to stronger separation and hence better MisD performance, but pushing $\lambda \to 0$ results in overly aggressive probability updates, causing instability and potential over-fitting on the calibration set. Therefore, in practice, an intermediate $\lambda$ is chosen to balance separation strength and stability.

**Property 2: Precision dominance over perfect calibration.** The proposed reliability curve guarantees strictly higher precision than perfect calibration for correctly classified samples in the high-confidence region, and symmetrically outperforms perfect calibration for misclassification detection in the low-confidence region. Consequently, given the distribution of samples over confidence values (*i.e.,* $w(s)$), we arrive at the following theorem (The proof is listed in the Appendix A.4):

**Theorem 4.4.** *Let* $r \in (0.5, 1)$ *and let* $w(s) \geq 0$ *be any weight function on* $[0, 1]$ *with* $\int_r^1 w(s)\,ds > 0$. *The precision for correct-prediction detection (*i.e., $\mathrm{Prec}^+$*) satisfies the following inequality:*

$$\mathrm{Prec}^+_{\Psi}(r; w) := \frac{\int_r^1 w(s)\,\Psi(s)\,ds}{\int_r^1 w(s)\,ds} \;\geq\; \frac{\int_r^1 w(s)\,s\,ds}{\int_r^1 w(s)\,ds} =: \mathrm{Prec}^+_{\mathrm{diag}}(r; w).$$

*Moreover, for the incorrect prediction detection with* $r \in (0, 0.5)$, *the precision (*i.e., $\mathrm{Prec}^-$*) satisfies the following inequality:*

$$\mathrm{Prec}^-_{\Psi}(r; w) := \frac{\int_0^r w(s)\,\left(1 - \Psi(s)\right)\,ds}{\int_0^r w(s)\,ds} \;\geq\; \frac{\int_0^r w(s)\,\left(1 - s\right)\,ds}{\int_0^r w(s)\,ds} =: \mathrm{Prec}^-_{\mathrm{diag}}(r; w).$$

**Property 3: Less tolerance for prediction mixing.** MisD emphasizes separating correctly and incorrectly classified samples; hence, less *mixing* between these two groups at the confidence level is critical. Entropy is a natural metric that can be used to measure the degree of mixing. Therefore, we can characterize the tolerance of the reliability curve for prediction mixing at every confidence level by the entropy. We then have the following theorem (The proof is listed in Appendix A.6):

**Theorem 4.5.** *Given arbitrary confidence* $s$, *then the corresponding entropy for the proposed reliability curve(*i.e., $\Psi(s)$*) and perfect-calibration curve satisfy the following inequality.*

$$\mathbf{M}_{\Psi}(s) \leq \mathbf{M}_{\mathrm{diag}}(s), \quad \mathbf{M}(s) := -\mathbb{P}(N_s^+) \log(\mathbb{P}(N_s^+)) - \mathbb{P}(N_s^-) \log(\mathbb{P}(N_s^-)), \tag{7}$$

*where* $\mathbb{P}(N_s^+)$ *and* $\mathbb{P}(N_s^-)$ *denote the probabilities of being correctly and incorrectly classified given confidence* $s$, *respectively;* i.e., $\mathbb{P}(N_s^+) = \mathbb{P}(\text{correct}|\text{conf} = s)$ *and* $\mathbb{P}(N_s^-) = 1 - \mathbb{P}(N_s^+)$.

This theorem indicates that the proposed reliability curve (*i.e.,* $\Psi(s)$) has less tolerance for mixing of correctly and incorrectly classified samples.

Taken together, **Properties 1–3** provide a coherent theoretical picture: the proposed curve continuously interpolates between perfect calibration and ideal separation (**Property 1**), provably achieves higher precision for both correct and incorrect prediction detection (**Property 2**), and admits lower entropy at each confidence level, indicating less tolerance for prediction mixing (**Property 3**). These results highlight that $\Psi(s)$ aligns better with the goal of improving MisD. Building on these insights, we next introduce a practical post-hoc method that operationalizes the proposed reliability curve.

## 4.2 SURROGATE LOSS

Despite the advantages of the proposed reliability curve (*i.e.,* $\Psi(s)$), directly realizing it as a training objective is challenging. In particular, the reliability curve cannot be straightforwardly expressed as a differentiable loss, and an alternative would be to approximate it through binning on a held-out calibration set, similar to ECE (Guo et al., 2017). However, calibration sets are usually small in practice, which causes large variance in the empirical bin estimates and makes such optimization unreliable. To tackle this issue, we propose a simple alternative to achieve it.

Specifically, we approximate the desired behavior of the reliability curve through a differentiable surrogate penalty. Recall our proposed curve $\Psi(s)$, which smoothly maps the confidence score $s \in [0, 1]$ to an expected accuracy curve, describing how prediction accuracy is expected to vary with confidence. Under this formulation, each prediction contributes an observed correctness signal. When a misclassified sample receives high confidence (e.g., $s$), its observed accuracy (0) lies far below the expected value $\Psi(s)$ on the sigmoid-shaped reliability curve at that confidence level. The

resulting gap therefore induces a penalty proportional to $\Psi(s) - 0$, reducing upward distortion in the high-confidence region of the reliability curve. Similarly, when a correctly classified sample is assigned low confidence, its observed accuracy (1) lies far above the expected value $\Psi(s)$ in the low-confidence region. The corresponding gap, $1 - \Psi(s)$, induces a penalty that encourages its confidence to increase toward the expected accuracy, thereby correcting distortion in the low-confidence region. However, $\Psi(s)$ only constrains the confidence, *i.e.,* the highest probability, while ignoring the distribution of the remaining probabilities. This may lead to optimization difficulties and suboptimal performance. Therefore, we further incorporate a constraint on the full probability distribution to make the constraint more comprehensive:

$$
\begin{aligned}
\mathcal{L}_{\mathrm{SUR}} = \min \beta \frac{1}{|N^+|} \sum_{i \in N^+} \phi\{1 - \Psi(s), -\mathbf{y}_i^\top \log f(\mathbf{x}_i)\} \\
+ (1 - \beta) \frac{1}{|N^-|} \sum_{j \in N^-} \phi\{\Psi(s), -[1/c, \dots, 1/c]^\top \log f(\mathbf{x}_j)\},
\end{aligned}
\tag{8}
$$

where $\beta$ is a hyperparameter balancing the constraints on correctly and incorrectly predicted samples, and $\phi\{\cdot, \cdot\}$ is a fusion function, which can be chosen as summation or multiplication. The first term of $\phi\{\cdot, \cdot\}$ encourages the reliability curve to align with the $\Psi(s)$. The second term of $\phi\{\cdot, \cdot\}$ provides a full probability constraint to avoid the limitation of only constraining the predicted class probability from the first term. Specifically, for correctly predicted samples, it is a standard Cross Entropy, which enforces a low-entropy probability distribution, while for incorrectly predicted samples, it imposes a constraint by comparing the predicted probability vector with the uniform distribution $[1/c, \dots, 1/c]$, thereby encouraging a high-entropy probability distribution. In summary, the first term drives the main alignment with the proposed reliability curve (*i.e.,* $\Psi(s)$), and the second term complements it with full-probability regularization, jointly yielding a more stable and effective objective.

### 4.3 LIGHTWEIGHT META NETWORK

To avoid altering the model predictions and introducing significant overhead, we propose a *post-hoc* method. It employs a lightweight meta network (*i.e.,* LMN) to learn the temperature coefficients $\tau$ for refining the VLM's confidence. For instance, given an image $v$, its logits can be refined as:

$$
\mathbf{z}'_v = [\tau_v \cdot z_{v,1}, \tau_v \cdot z_{v,2}, \dots, \tau_v \cdot z_{v,c}], \quad z_{v,i} = \tau_{\mathrm{clip}} \cdot \mathrm{sim}(\xi(\mathbf{x}_v), \psi(\mathbf{t}_i)).
\tag{9}
$$

To obtain the instance-specific temperature coefficient $\tau_v$, we design a lightweight network that takes as input the information from both modalities, without modifying the parameters of the pretrained or fine-tuned model. Concretely, the model's output logits $\mathbf{z}_v$, the image embedding $\xi(\mathbf{x}_v)$, and the predicted text embeddings $\psi(t_p)$ are each passed through a separate fully connected (FC) layer to map them into a common latent space. These representations are then concatenated and fed into another FC layer, which projects them to a scalar value corresponding to $\tau_v$. Formally, we have:

$$
\tau_v = \sigma^+\big(\mathrm{FC}_\tau(\mathbf{h}_z || \mathbf{h}_x || \mathbf{h}_t)\big), \quad \mathbf{h}_z = \mathrm{FC}_z(\mathbf{z}_v), \quad \mathbf{h}_x = \mathrm{FC}_x(\xi(\mathbf{x}_v)), \quad \mathbf{h}_t = \mathrm{FC}_t(\psi(\mathbf{t}_p)),
\tag{10}
$$

where $[\cdot || \cdot]$ denotes concatenation and $\sigma^+(x) = \log(1 + \exp(x))$ is an element-wise softplus activation (Dugas et al., 2000). Finally, by substituting Eq.(9) into the objective function in Eq.(8), we update only the parameters of these FC layers on the calibration set, making the overall training procedure lightweight and efficient. The flowchart of the proposed meta network can be found in Figure 5.

## 5 EXPERIMENT

In this section, we conduct experiments on six public datasets to evaluate the proposed method in terms of different settings. Details of experiments are shown in Appendix C and additional experiments are shown in Appendix D. The code is released at Code Link.

### 5.1 EXPERIMENTAL SETUP

**Datasets.** We conduct the analysis on six datasets covering specialized and fine-grained domains, which include DTD (Cimpoi et al., 2014), Flowers102 (Nilsback & Zisserman, 2008), EuroSAT (Helber et al., 2019), RESICS45 (Cheng et al., 2017), MNIST (Deng, 2012), and CUB (Wah et al.,

Table 1: Comparison results of recent confidence calibration OOD detection methods in the few-shot setting. Note that ↑ indicates higher is better, ↓ indicates lower is better.

| | DTD | | | | | Flowers102 | | | | |
| Methods | AUROC↑ | AUPR-S↑ | AUPR-E↑ | FPR90-S↓ | FPR90-E↓ | AUROC↑ | AUPR-S↑ | AUPR-E↑ | FPR90-S↓ | FPR90-E↓ |
|---|---|---|---|---|---|---|---|---|---|---|
| Zero-shot CLIP | 0.762 | 0.740 | 0.771 | 0.669 | 0.572 | 0.864 | 0.922 | 0.759 | 0.435 | 0.354 |
| FeatureClipping | 0.749 | 0.716 | 0.764 | 0.687 | 0.571 | 0.873 | 0.935 | 0.695 | 0.416 | 0.321 |
| SCT | 0.759 | 0.741 | 0.776 | 0.685 | 0.557 | 0.868 | 0.926 | 0.761 | 0.429 | 0.337 |
| ViLU | 0.769 | 0.759 | 0.762 | 0.678 | 0.521 | 0.875 | 0.913 | 0.772 | 0.401 | 0.329 |
| LMN (Ours) | **0.802** | **0.800** | **0.804** | **0.636** | **0.457** | **0.886** | **0.937** | **0.799** | **0.378** | **0.305** |

| | EuroSAT | | | | | RESICS45 | | | | |
| Methods | AUROC↑ | AUPR-S↑ | AUPR-E↑ | FPR90-S↓ | FPR90-E↓ | AUROC↑ | AUPR-S↑ | AUPR-E↑ | FPR90-S↓ | FPR90-E↓ |
|---|---|---|---|---|---|---|---|---|---|---|
| Zero-shot CLIP | 0.65 | 0.501 | 0.771 | 0.782 | 0.742 | 0.779 | 0.824 | 0.711 | 0.636 | 0.508 |
| FeatureClipping | 0.685 | 0.613 | 0.727 | 0.729 | 0.536 | 0.781 | 0.827 | 0.705 | 0.638 | 0.501 |
| SCT | 0.681 | 0.534 | 0.791 | 0.754 | 0.682 | 0.784 | 0.826 | 0.716 | 0.633 | 0.501 |
| ViLU | 0.723 | 0.618 | 0.787 | 0.723 | 0.538 | 0.787 | 0.829 | 0.730 | 0.618 | 0.493 |
| LMN (Ours) | **0.788** | **0.698** | **0.855** | **0.655** | **0.468** | **0.808** | **0.845** | **0.741** | **0.597** | **0.445** |

| | MNIST | | | | | CUB | | | | |
| Methods | AUROC↑ | AUPR-S↑ | AUPR-E↑ | FPR90-S↓ | FPR90-E↓ | AUROC↑ | AUPR-S↑ | AUPR-E↑ | FPR90-S↓ | FPR90-E↓ |
|---|---|---|---|---|---|---|---|---|---|---|
| Zero-shot CLIP | 0.813 | 0.565 | 0.919 | 0.511 | 0.482 | 0.807 | 0.839 | 0.758 | 0.767 | 0.554 |
| FeatureClipping | 0.816 | 0.654 | 0.843 | 0.501 | 0.461 | 0.805 | 0.834 | 0.756 | 0.768 | 0.541 |
| SCT | 0.837 | 0.664 | 0.933 | 0.486 | 0.423 | 0.808 | 0.839 | 0.759 | 0.766 | 0.547 |
| ViLU | 0.877 | 0.769 | 0.954 | 0.350 | 0.263 | 0.801 | 0.827 | 0.753 | 0.769 | 0.563 |
| LMN (Ours) | **0.915** | **0.779** | **0.965** | **0.200** | **0.205** | **0.812** | **0.846** | **0.764** | **0.756** | **0.532** |

2011). For each dataset, we adopt the official training and test splits provided in Li et al. (2022), and construct a few-shot calibration set by randomly sampling a small subset from the training set.

**Baselines.** We evaluate our method using pretrained CLIP as the base model and compare it with two recent calibration approaches (FeatureClipping (Tao et al., 2025) and DOR (Wang et al., 2025)), a strong VLM-based OOD detector (SCT (Yu et al., 2024)), and the latest MisD-oriented method for VLMs (ViLU (Lafon et al., 2025)). We also acknowledge FSMisD (Zeng et al., 2025), a recent MisD-oriented method for VLMs, but its code is not publicly available and cannot be included. In addition, we evaluate two commonly used prompt-tuning CLIP variants: CoOP (Zhou et al., 2022b) (textual) and VPT (Jia et al., 2022) (visual).

**Implementation details.** We use CLIP ViT-B/32 as the visual backbone. For both visual and textual prompt learning, we set the prefix size to 16 (Zhou et al., 2022b; Jia et al., 2022). We use SGD as the optimizer, with the number of training epochs selected from $\{100, 150, 200\}$ and the learning rate selected from $\{0.001, 0.002, 0.005\}$. We adopt a 16-shot setting for the calibration set. When the base model is prompt-tuning CLIP, we split the original 16-shot calibration set into two parts: one is used for learning the prompts, and the other is reserved for training our post-hoc calibration network.

**Evaluation metrics.** To evaluate the MisD, we measure the ranking capability of confidence scores, *i.e.,* the ability to rank correctly classified samples ahead of misclassified ones. Following existing works (Corbière et al., 2019; Hendrycks et al., 2018), we adopt several widely used metrics: AUROC, AUPR-Success, AUPR-Error, FPR90%-Success-TPR, and FPR90%-Error-TPR. Since our MisD evaluation simultaneously considers both correct and incorrect predictions, the positive class in AUPR and FPR is redefined according to the detection target: for *Success*, correctly predicted samples are treated as positives, whereas for *Error*, incorrectly predicted samples are treated as positives.

## 5.2 MAIN RESULTS

We first evaluate the MisD performance of recent confidence calibration methods, the OOD detection method, and our proposed method applied to the pretrained CLIP. The results on five commonly used metrics (*i.e.,* AUROC, AUPR-Success, AUPR-Error, FPR90-Success, and FPR90-Error) are reported in Table 1, from which we make the following observations:

First, *the confidence calibration methods cannot help CLIP to distinguish between the correctly predicted samples and the incorrectly predicted samples*. For example, on average across all datasets, FeatureClipping improves AUROC by only about 0.7% over the pretrained CLIP. Moreover, such marginal improvements are further undermined by their inconsistency, as both methods fail to enhance CLIP uniformly across all evaluation metrics. This aligns with Theorem 4.2, confirming that confidence calibration imposes a fundamental limit on achievable MisD performance.

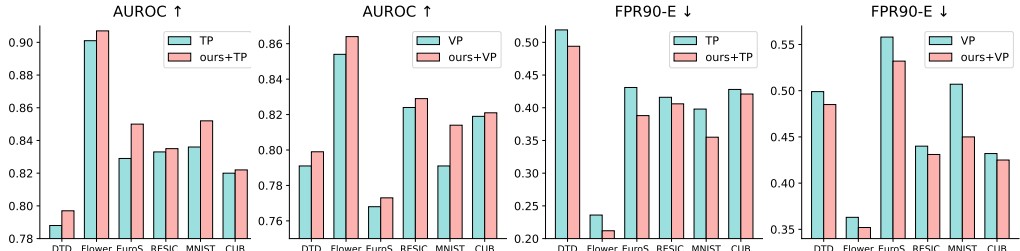

Figure 3: Bar plot comparison of textual-prompt (*i.e.,* TP in Figure) and visual-prompt (*i.e.,* VP in Figure) CLIP before and after applying our proposed method, evaluated in terms of AUROC and FPR90-Error across DTD, Flowers102, EuroSAT, RESICS45, MNIST, and CUB datasets.

Second, compared to the VLM-based calibration and OOD detection methods, *the proposed method always outperforms them by large margins across all evaluation metrics.* For example, the proposed method on average improves by 6.1%, 10.5%, 4.7%, 13.4%, and 22.9%, compared to the pretrained CLIP, in terms of AUROC, AUPR-Success, AUPR-Error, FPR90-Success, and FPR90-Error. Furthermore, even when compared with the strongest baseline ViLU, our method still achieves substantial gains of 2.8%, 5.1%, 2.1%, 5.4%, and 18.7% on the same metrics. These results demonstrate the superiority of the proposed method.

## 5.3 EVALUATION ON VARIOUS BASE MODELS

To evaluate the effectiveness of the proposed method on more base models, we adopt two types of prompt-based CLIP models (*i.e.,* textual prompt and visual prompt). Since prompt-based methods require a training set to learn the prompt embeddings, we leverage the few-shot calibration set, which is originally divided into training and validation portions. Specifically, following (Wang et al., 2021), the training portion is used to learn the prompt embeddings, while the validation portion serves both to validate the learned prompts and to train the proposed post-hoc calibration model.

The results are shown in Figure 3. We have the following observations: our method consistently improves the performance of both types of prompt-based CLIP across multiple datasets. For example, on average, our method brings an improvement of 1.2% in AUROC and a reduction of 6.2% in FPR90-Error for the textual prompt, and 1.1% and 4.4%, respectively, for the visual prompt. These results demonstrate the applicability of our model to different forms of base models.

## 5.4 EVALUATION ON OPEN-VOCABULARY SETTING

A key advantage of pretrained CLIP lies in its strong zero-shot capability, enabling it to generalize well to unseen classes. However, fine-tuning operations, including prompt learning, may compromise this ability by overfitting to the training classes. To examine whether our proposed post-hoc method preserves the zero-shot capability, we design an open-vocabulary evaluation and compare LMN with both zero-shot CLIP and the open-vocabulary calibration method DOR (Wang et al., 2025) (a CoOP-based calibration method). Specifically, we randomly sample a subset of classes for calibration, while the remaining unseen classes are reserved for testing. The results are shown in Table 2.

The experimental results demonstrate that our method not only avoids degrading CLIP's performance under the open-vocabulary setting, but also leads to consistent improvements across almost all datasets and evaluation metrics. For example, on the DTD dataset, our method achieves a 7.1% and 5.9% relative improvement over CLIP and DOR in terms of the FPR90-S metric. This indicates that our approach is capable of enhancing model performance while preserving CLIP's inherent strengths. We attribute this to the post-hoc nature of our method, which does not modify the pretrained parameters of CLIP and thus maintains the intrinsic generalization ability of its image and text embeddings. By retaining this property, the proposed calibration method can leverage the strong representational power of CLIP while improving its reliability on unseen classes.

Table 2: Results of open-vocabulary evaluation on six datasets, measured by AUROC, FPR90-S, and FPR90-E. ↑ denotes that higher values are better, while ↓ denotes that lower values are better.

| Methods | DTD | | | Flowers102 | | | EuroSAT | | |
|---|---|---|---|---|---|---|---|---|---|
| | AUROC↑ | FPR90-S↓ | FPR90-E↓ | AUROC↑ | FPR90-S↓ | FPR90-E↓ | AUROC↑ | FPR90-S↓ | FPR90-E↓ |
| Zero-shot CLIP | 0.760 | 0.642 | 0.604 | 0.853 | 0.407 | 0.397 | 0.608 | 0.747 | 0.882 |
| DOR | 0.758 | 0.631 | 0.612 | 0.847 | 0.410 | 0.394 | 0.605 | 0.751 | 0.883 |
| LMN (Ours) | **0.770** | **0.596** | **0.598** | **0.858** | **0.396** | **0.387** | **0.614** | **0.743** | **0.875** |

| Methods | RESICS45 | | | MNIST | | | CUB | | |
|---|---|---|---|---|---|---|---|---|---|
| | AUROC↑ | FPR90-S↓ | FPR90-E↓ | AUROC↑ | FPR90-S↓ | FPR90-E↓ | AUROC↑ | FPR90-S↓ | FPR90-E↓ |
| Zero-shot CLIP | 0.780 | 0.601 | 0.536 | 0.854 | 0.545 | 0.322 | 0.789 | **0.641** | 0.490 |
| DOR | 0.764 | 0.621 | 0.543 | 0.842 | 0.573 | 0.331 | 0.779 | 0.667 | 0.512 |
| LMN (Ours) | **0.782** | **0.599** | **0.527** | **0.859** | **0.539** | **0.301** | **0.790** | 0.653 | **0.472** |

Figure 4: Bar plot comparison of AUROC, FPR90-S, and FPR90-E metrics of CLIP, CLIP with full probabilities constraint, and CLIP with our full method (*i.e.,* first three figures) on the RESICS45 dataset. The corresponding reliability diagrams are shown in the last three figures.

## 5.5 ABLATION STUDIES

The key component of the proposed method is the surrogate loss $\mathcal{L}_{\mathrm{SUR}}$, which consists of two parts: (i) a confidence regularization term that encourages the predictions to align with the target reliability curve (*i.e.,* $\Phi(s)$), and (ii) a full-probability constraint (FPC) that complements the confidence regularization by regularizing the entire probability distribution, preventing the training instability cause by single probability (*i.e.,* confidence) were optimized. Therefore, we do not report a separate ablation with only the confidence regularization term, as it is intended to be used together with FPC. To verify the effectiveness of these components, we visualize the reliability diagrams on the RESICS45 dataset under three settings: the original CLIP, CLIP with FPC, and CLIP with the full proposed method. The visualizations together with the corresponding quantitative results are reported in Figure 4. Additional ablation studies on other datasets are provided in Appendix D.1.

From Figure 4, we draw the following observations. First, the complete objective achieves the best overall performance, and using FPC alone also improves MisD performance compared to the pretrained CLIP, indicating that both components of the surrogate loss contribute effectively. Second, from the reliability diagrams, we see that the pretrained CLIP model exhibits overconfidence. Incorporating FPC alleviates this issue to some extent; however, the resulting curve still deviates from the target $\Phi(s)$. Finally, with the proposed confidence regularization term, the reliability curve aligns much more closely with $\Phi(s)$, demonstrating that our surrogate loss indeed possesses the desired ability to calibrate predictions toward the target normalized sigmoid reliability curve.

## 6 CONCLUSION

In this work, we revisited the reliability diagram of confidence calibration and established its connection with MisD. We showed that the standard calibration objective inherently limits MisD performance, and proposed a new reliability curve as the calibration objective. This reliability curve theoretically improves MisD performance and reduces the mixing of correct and incorrect predictions. As the value of the reliability curve is a statistical result and thus non-differentiable, we address this issue by designing a differentiable surrogate loss. Furthermore, to preserve the predictive power of VLMs, we developed a lightweight post-hoc framework that employs a meta network to produce sample-specific temperature factors. Both theoretical analysis and extensive experiments confirmed that our approach consistently enhances MisD performance while maintaining model accuracy.

## ACKNOWLEDGEMENT

Xiaofeng Zhu was supported in part by the National Key Research and Development Program of China under Grant (No.2022YFA1004100). Ping Hu was supported in part by the National Natural Science Foundation of China under Grant (No.62476048) and the Sichuan Science and Technology Program under Grant (No.2025YFMS0004). Xiaoshuang Shi was supported in part by the National Natural Science Foundation of China (No. 62276052).

## REPRODUCIBILITY STATEMENT

We have made extensive efforts to ensure the reproducibility of our work. The model architecture, training procedure, and hyperparameter settings are described in Appendix C.2. Complete proofs of the theoretical results are presented in Appendix A. The datasets used in our experiments are publicly available, and the preprocessing steps are explained in the Appendix C D. Moreover, we provide an anonymous link to the source code in the supplementary materials to facilitate reproduction of our experiments (Code Link).

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

# A THEORETICAL PROOF

## A.1 PROOF FOR LEMMA 4.1

**Lemma A.1.** *Given a confidence interval $[a, b]$, let $w(s)$ denote the density of the sample at confidence level $s$. Then, the precision of correct predictions within $[a, b]$ can be derived from region under the reliability curve $f(s)$ over $[a, b]$, i.e., $\text{Prec}^+ = (\int_a^b w(s)f(s)ds)/\int_a^b w(s)ds$. Similarly, the precision of incorrect predictions over $[a, b]$ can be derived from the region above the curve via $\text{Prec}^- = (\int_a^b w(s)(1 - f(s))ds)/\int_a^b w(s)ds$.*

*Proof.* The formula of precision is defined as:

$$\text{Precision}_{[a,b]} = \frac{\text{correct predictions with } s \in [a, b]}{\text{all predictions with } s \in [a, b]}. \tag{11}$$

The reliability curve can be represented as $f(s) = \mathbb{P}(\text{correct}|\text{confidence} = s)$, thus the number of correct predictions in the interval $[a, b]$ can be represented as:

$$\text{correct predictions} = \int_a^b w(s)\text{Accuracy}(s)ds = \int_a^b w(s)\mathbb{P}(\text{correct}|\text{confidence} = s)ds. \tag{12}$$

The number of the whole sample is:

$$\text{all predictions} = \int_a^b w(s)ds. \tag{13}$$

Combining the above three equations, we have the following equation:

$$\text{Prec}_{[a,b]}^+ = \frac{\int_a^b w(s)\mathbb{P}(\text{correct}|\text{confidence} = s)ds}{\int_a^b w(s)ds} = \frac{\int_a^b w(s)f(s)ds}{\int_a^b w(s)ds}, \tag{14}$$

where $\int_a^b \mathbb{P}(\text{correct}|\text{confidence} = s)ds$ is the area under the reliability curve over a given confidence interval $[a, b]$. Then the precision equals that area divided by the length of the interval.

For the incorrect prediction detection, which just needs to replace the numerator of Eq. (11) with the number of incorrect predictions, thus we have:

$$\text{Prec}_{[a,b]}^- = \frac{\int_a^b w(s)\mathbb{P}(\text{incorrect}|\text{confidence} = s)ds}{\int_a^b w(s)ds} = \frac{\int_a^b w(s)(1 - f(s))ds}{\int_a^b w(s)ds}. \tag{15}$$

The proof is completed.

$\square$

## A.2 PROOF FOR COROLLARY 4.3

**Corollary A.2** (Ideal separation case). *When $\lambda \to 0$, the proposed normalized sigmoid curve assigns higher confidence to all correctly classified samples than to any misclassified sample, i.e.,*

$$\mathbb{P}\big(\text{Conf}(N^+) > \text{Conf}(N^-)\big) = 1, \tag{16}$$

*where $N^+$ and $N^-$ denote the sets of correctly classified and misclassified samples, respectively. In this case, the reliability curve approaches a step-like curve (cf. Fig. 2 with $\lambda = 1e - 5$).*

*Proof.* As $\lambda \to 0$, Figure 2 shows that all samples with confidence greater than 0.5 are correctly classified (accuracy 100%), while all samples with confidence below 0.5 are misclassified (accuracy 0%). This implies that the confidence of every correctly classified sample is strictly greater than 0.5, and the confidence of every misclassified sample is strictly less than 0.5. Therefore, the confidence of any correctly classified sample is always higher than that of any misclassified sample. $\square$

## A.3 PROOF FOR THEOREM 4.2

**Theorem A.3.** *Let $r \in [0,1]$ be a confidence threshold. Under perfect calibration, the precision for the correct prediction detection and incorrect prediction detection tasks is $\mathrm{Prec}_{[r,1]}^{+} = \mathbb{E}_{s \sim w(s|s \in [r,1])}[s]$ and $\mathrm{Prec}_{[0,r]}^{-} = \mathbb{E}_{s \sim w(s|s \in [0,r])}[1-s]$, respectively.*

*Proof.* Given a perfect calibration model, which reliability curve is a diagonal line (*i.e.*, $f(s) = s$). For the correct prediction detection task, predictions with confidence above the threshold $r$ (*i.e.*, $s \in [r,1]$) are considered as detected correct predictions. By Lemma 4.1, the precision of correct prediction in $[r,1]$ is

$$\mathrm{Prec}_{[r,1]}^{+} = \frac{\int_r^1 w(s)f(s)ds}{\int_r^1 w(s)ds}. \tag{17}$$

Substituting $f(s) = s$, we have:

$$\mathrm{Prec}_{r,1}^{+} = \frac{\int_r^1 w(s)f(s)ds}{\int_r^1 w(s)ds} = \mathbb{E}_{s \sim w(s|s \in [r,1])}[s]. \tag{18}$$

Similarly, the precision of incorrect predictions is:

$$\mathrm{Prec}_{[0,1=r]}^{-} = \frac{\int_0^r w(s)(1-f(s))ds}{\int_0^r w(s)ds} = \mathbb{E}_{s \sim w(s|s \in [0,r])}[1-s]. \tag{19}$$

The proof is completed. $\qquad\square$

## A.4 PROOF FOR THEOREM 4.4

**Theorem A.4** (Theorem 4.4 (restated)). *Let $r \in (0.5, 1)$ and let $w \geq 0$ be any weight function on $[0,1]$ with $\int_r^1 w(s)\,ds > 0$. The precision for correct-prediction detection (*i.e.*, $\mathrm{Prec}^{+}$) satisfies the following inequality:*

$$\mathrm{Prec}_{\Psi}^{+}(r;w) := \frac{\int_r^1 w(s)\,\Psi(s)\,ds}{\int_r^1 w(s)\,ds} \geq \frac{\int_r^1 w(s)\,s\,ds}{\int_r^1 w(s)\,ds} =: \mathrm{Prec}_{\mathrm{diag}}^{+}(r;w).$$

*Moreover, for the incorrect prediction detection with $r \in (0, 0.5)$, the precision (*i.e.*, $\mathrm{Prec}^{-}$) satisfies the following inequality:*

$$\mathrm{Prec}_{\Psi}^{-}(r;w) := \frac{\int_0^r w(s)\left(1-\Psi(s)\right)ds}{\int_0^r w(s)\,ds} \geq \frac{\int_0^r w(s)\left(1-s\right)ds}{\int_0^r w(s)\,ds} =: \mathrm{Prec}_{\mathrm{diag}}^{-}(r;w).$$

*Proof.* Before proving the Theorem, we state the following lemma (proof in Appendix A.5).

**Lemma A.5** (Above-diagonal on the high-confidence side). *Given the proposed normalized sigmoid curve (*i.e.*, $\Psi(x)$) and diagonal line (*i.e.*, $y = x$), then $\Psi$ is strictly concave on $[0.5, 1]$, satisfies $\Psi(0.5) = 0.5$ and $\Psi(1) = 1$, and hence:*

$$\Psi(x) \geq x \quad \text{for all } x \in [0.5, 1], \qquad \Psi(x) > x \quad \text{for all } x \in (0.5, 1).$$

By Lemma A.5, let $D(s) := \Psi(s) - s \geq 0$ for $s \in [r,1]$ and $D(s) > 0$ on $(r,1)$. Therefore

$$\int_r^1 w(s)\,\Psi(s)\,ds - \int_r^1 w(s)\,s\,ds = \int_r^1 w(s)\,D(s)\,ds \geq 0,$$

and the inequality is strict if $w$ places positive mass on some subset of $(r,1)$ where $D > 0$. Division by the common normalizer $\int_r^1 w(s)\,ds > 0$ yields the claim, then the inequality $\mathrm{Prec}_{\Psi}^{+}(r;w) > \mathrm{Prec}_{\mathrm{diag}}^{+}$ holds.

For the incorrect prediction detection, since $\Psi(s) \leq s$ on $[0, 0.5]$ (strict on $(0, 0.5)$ by symmetry of the above argument), an analogous statement holds for incorrect prediction detection on $[0, r]$ with $r < 0.5$: for any nonnegative $w$ with $\int_0^r w(s)\, ds > 0$,

$$\frac{\int_0^r w(s)\,(1 - \Psi(s))\, ds}{\int_0^r w(s)\, ds} \geq \frac{\int_0^r w(s)\,(1 - s)\, ds}{\int_0^r w(s)\, ds}.$$

Then the inequality $\mathrm{Prec}_\Psi^-(r; w) > \mathrm{Prec}_{\mathrm{diag}}^-$ holds.

$\square$

## A.5 PROOF FOR LEMMA A.5

**Lemma** (Lemma A.5 (restated))**.** *Given the proposed normalized sigmoid curve (i.e., $\Psi(x)$) and diagonal line (i.e., $y = x$), then $\Psi$ is strictly concave on $[0.5, 1]$, satisfies $\Psi(0.5) = 0.5$ and $\Psi(1) = 1$, and hence:*

$$\Psi(x) \geq x \quad \text{for all } x \in [0.5, 1], \qquad \Psi(x) > x \quad \text{for all } x \in (0.5, 1).$$

*Proof.* Let $\phi(x) := \sigma\left(\frac{x - 0.5}{\lambda}\right)$. Since $\sigma'(z) = \sigma(z)\left(1 - \sigma(z)\right)$ and $\sigma''(z) = \sigma(z)\left(1 - \sigma(z)\right)\left(1 - 2\sigma(z)\right)$, we have for $x > 0.5$ that $\frac{x - 0.5}{\lambda} > 0$ and thus $\sigma\left(\frac{x - 0.5}{\lambda}\right) > \frac{1}{2}$, implying $\sigma''\left(\frac{x - 0.5}{\lambda}\right) < 0$. Hence

$$\phi''(x) = \frac{1}{\lambda^2}\sigma''\left(\frac{x - 0.5}{\lambda}\right) < 0 \quad (x \in (0.5, 1)),$$

i.e., $\phi$ is strictly concave on $[0.5, 1]$. The normalization

$$\Psi(x) = \frac{\phi(x) - \phi(0)}{\phi(1) - \phi(0)}, \qquad \phi(0) = \sigma\left(-\frac{0.5}{\lambda}\right), \quad \phi(1) = \sigma\left(\frac{0.5}{\lambda}\right),$$

is an affine transform and therefore preserves strict concavity. Direct evaluation gives

$$\Psi(1) = 1, \qquad \Psi(0.5) = \frac{\sigma(0) - \sigma(-a)}{\sigma(a) - \sigma(-a)} = \frac{\frac{1}{2} - (1 - \sigma(a))}{\sigma(a) - (1 - \sigma(a))} = \frac{1}{2}, \qquad a = \frac{0.5}{\lambda} > 0.$$

Define $F(x) := \Psi(x) - x$. Since $\Psi$ is strictly concave and $-x$ is linear (hence both concave and convex), $F$ is strictly concave on $[0.5, 1]$, and $F(0.5) = F(1) = 0$. For any $x \in (0.5, 1)$, write $x = \lambda \cdot 0.5 + (1 - \lambda) \cdot 1$ with $\lambda \in (0, 1)$; strict concavity yields

$$F(x) > \lambda F(0.5) + (1 - \lambda)F(1) = 0.$$

Thus $\Psi(x) > x$ on $(0.5, 1)$ and $\Psi(x) \geq x$ on $[0.5, 1]$.

$\square$

## A.6 PROOF FOR THEOREM 4.5

**Theorem A.6** (Theorem 4.5 (restated))**.** *Given arbitrary confidence $s$, then the corresponding entropy for the proposed reliability curve(i.e., $\Psi(s)$) and perfect-calibration curve satisfy the following inequality.*

$$\mathbf{M}_\Psi(s) \leq \mathbf{M}_{\mathrm{diag}}(s), \quad \mathbf{M}(s) := -\mathbb{P}(N_s^+)\log(\mathbb{P}(N_s^+)) - \mathbb{P}(N_s^-)\log(\mathbb{P}(N_s^-)), \tag{20}$$

*where $N_s^+$ and $N_s^-$ denote the conditional probabilities of being correctly and incorrectly classified given confidence $s$, respectively; i.e., $N_s^+ = \mathbb{P}(\text{correct}|\text{conf} = s)$ and $N_s^- = 1 - N_s^+$.*

*Proof.* Since the vertical axis of the reliability diagram is accuracy, and given an arbitrary confidence $s$, we have $\mathbb{P}(N_s^+) = \mathrm{Acc}(s)$ and $\mathbb{P}(N_s^-) = 1 - \mathrm{Acc}(s)$. Due to $\mathbb{P}(N_s^+) + \mathbb{P}(N_s^-) = 1$, then the entropy can written as binary entropy form:

$$\mathbf{M}(s) = -(p_s \ln p_s + (1 - p_s)\ln(1 - p_s)), \tag{21}$$

where $p_s = \mathbb{P}(N_s^+)$.

Then, we can obtain the first-order derivation of $p_s$:

$$\frac{\partial \mathbf{M}(s)}{\partial p_s} = -(\ln p_s + 1 - (\ln(1-p) + 1)) = \ln \frac{1-p}{p}. \tag{22}$$

When $p_s = 0.5$, $\mathbf{M}(s)$ reaches its maximum value in the interval $[0,1]$ and $\frac{\partial^2 \mathbf{M}(s)}{\partial p_s^2} < 0$. Thus, the closer $p_s$ is to 0.5, the larger the entropy becomes.

Let us recall the Lemma A.5, when $s \in [0, 0.5]$ we have $\Psi(s) \leq s \leq 0.5$, then:

$$|\Psi(s) - 0.5| = 0.5 - \Psi(s) \geq 0.5 - s = |s - 0.5|. \tag{23}$$

Similarly, we can easily obtain that when $s \in [0.5, 1]$ we have $\Psi(s) \geq s \geq 0.5$, then:

$$|\Psi(s) - 0.5| = \Psi(s) - 0.5 \geq s - 0.5 = |s - 0.5|. \tag{24}$$

Based on Eq. (23) and Eq. (24), $|\Psi(s) - 0.5| \geq |s - 0.5|$ holds for all $s \in [0, 1]$. Therefore, the $p_s$ of the perfect-calibration curve $y = x$ lies closer to 0.5 than that of the proposed $\Psi(s)$. As a result, the perfect calibration curve exhibits higher entropy, thereby tolerating a greater degree of mixing between correctly and incorrectly predicted samples.

$\square$

## B    MODEL DETAILS

### B.1    FLOWCHART VISUALIZATION

This section gives a flowchart (*i.e.,* Figure 5) of the proposed lightweight meta networks (LMN).

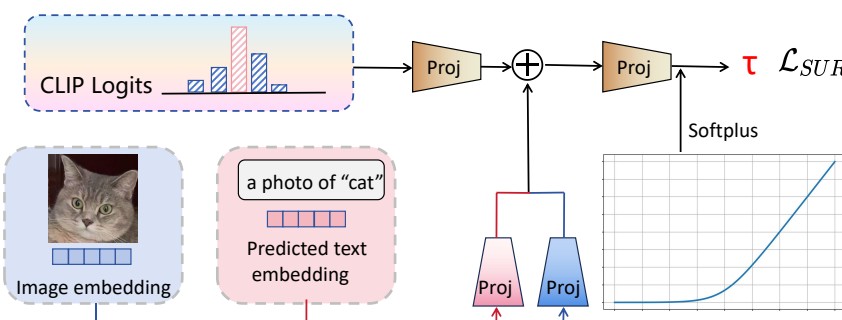

Figure 5:   Flowchart of the proposed lightweight meta network (LMN). Given the output logits of CLIP, image embedding, and predicted text embedding, LMN first projects them into a shared subspace using distinct FC layers (indicated by different colors in the figure). The resulting vectors are concatenated and mapped to a scalar via another FC layer, followed by a Softplus activation to produce the sample-specific $\tau$. Finally, the surrogate loss $\mathcal{L}_{\text{SUR}}$ updates the FC layers.

### B.2    UNDERSTANDING THE CONTRIBUTION OF EACH MODALITY IN LMN

The meta-network indeed leverages three types of signals (i.e., logits, image embeddings, and predicted text embeddings) and each contributes complementary information for predicting the effective temperature factor.

**Logits.**   Logits as the primary calibration signal.  Logits provide the most direct evidence for confidence misalignment, as they encode the inter-class margins and the overall shape of the predictive distribution. This is the core quantity used by traditional temperature scaling. Here we describe the complementary roles of each modality and why all of them contribute meaningfully to MisD:

**Image embeddings.** Image embeddings capture sample difficulty. Even when two samples share similar logits, their underlying visual characteristics may differ substantially. Image embeddings help

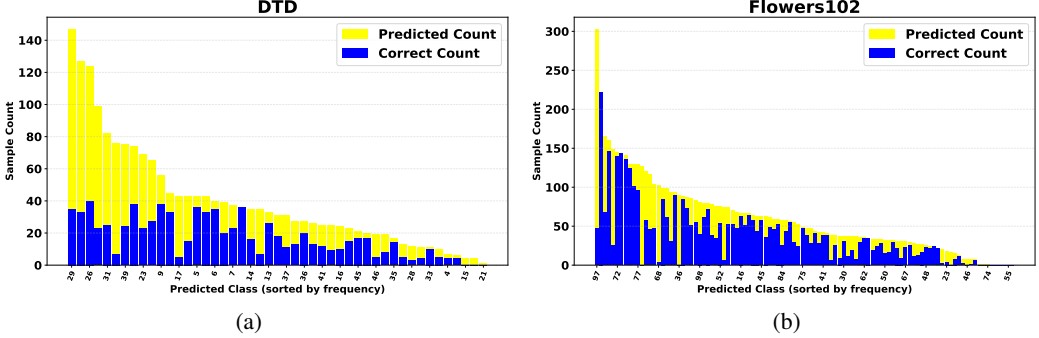

Figure 6: Predicted count (yellow) and correct count (blue) for each predicted class, sorted by prediction frequency.

LMN identify hard or visually atypical samples (e.g., unusual textures, crowded scenes, rare visual patterns). This allows LMN to incorporate sample-level visual cues.

**Predicted text embeddings.** Predicted text embeddings reveal systematic category-level patterns. To further illustrate the semantic confusion patterns captured by the predicted text embeddings, we visualize the distribution of predicted samples across classes in Figure 6. The figure shows that CLIP tends to cluster certain misclassified samples around a few semantically related text prototypes. Incorporating text embeddings enables LMN to capture these category-level tendencies and selectively increase correction strength for categories prone to systematic confusion.

Taken together, the three modalities provide complementary and non-redundant information: logits characterize confidence geometry, image embeddings characterize sample difficulty, and text embeddings capture semantic misalignment patterns.

### B.3 MOTIVATION FOR CHOOSING R

We set the midpoint parameter $r = 0.5$ because it corresponds to the natural neutral threshold separating "uncertain" from "confident" predictions, which aligns with standard confidence-based calibration intuition (Johansson et al., 2023). Moreover, placing the inflection point at the center of the interval $[0, 1]$ avoids introducing asymmetry or favoring either low- or high-confidence regions, providing a balanced and unbiased target curve.

## C EXPERIMENTAL SETTING

### C.1 DATASETS DETAILS

Table 3: The statistics of the used datasets.

|  | Num. classes ($|\mathcal{Y}|$) | Size training data | Avg. labeled data per class | Size test |
|---|---|---|---|---|
| DTD | 47 | 3760 | 64 | 1880 |
| Flowers102 | 102 | 2040 | 16 | 6149 |
| EuroSAT | 10 | 27000 | 2200 | 5000 |
| RESICS45 | 45 | 6300 | 110 | 25200 |
| MNIST | 10 | 60000 | 4696 | 10000 |
| CUB | 200 | 5594 | 26 | 5794 |

In the experiment section, we use six datasets. The statistics of the used datasets are reported in the Table 3. Here, we provide a description of each of them:

1. **DTD** (Cimpoi et al., 2014) The Describable Textures Dataset (DTD) is a continuously expanding collection of texture images captured in unconstrained environments. The

annotations are based on human-interpretable attributes, reflecting perceptual characteristics of textures.

2. **Flowers102** (Nilsback & Zisserman, 2008) It is a benchmark dataset that collects images of 102 distinct flower categories, each representing species commonly occurring in the United Kingdom.

3. **EuroSAT** (Helber et al., 2019) EuroSAT focuses on the classification of satellite scene images, providing 10 predefined categories for evaluation.

4. **RESICS45** (Cheng et al., 2017) Designed for remote sensing image scene classification, this benchmark dataset provides 45 categories of scenes and is publicly accessible.

5. **MNIST** (Deng, 2012) MNIST serves as a benchmark collection of handwritten digit images, in which all samples are normalized in terms of size and aligned to the center of a uniform image grid.

6. **CUB** (Wah et al., 2011) CUB is a fine-grained bird species classification dataset, containing 200 categories with large intra-class variation and subtle inter-class differences. It serves as a benchmark for subordinate categorization tasks and provides comprehensive annotations to support research in fine-grained visual recognition.

Table 4: Settings for the proposed method.

|  | learning rate | $\beta$ | Num. hidden | $\lambda$ | $\phi$ |
|---|---|---|---|---|---|
| DTD | 0.002 | 0.6 | 16 | 0.05 | summation |
| Flowers102 | 0.001 | 0.8 | 16 | 0.05 | summation |
| EuroSAT | 0.002 | 0.6 | 16 | 0.1 | summation |
| RESICS45 | 0.001 | 0.8 | 64 | 0.05 | multiply |
| MNIST | 0.005 | 0.4 | 16 | 0.05 | summation |
| CUB | 0.002 | 0.9 | 8 | 0.05 | multiply |

## C.2 IMPLEMENTATION DETAILS

We use CLIP ViT-B/32 as the visual backbone and report results averaged over 3 runs. For both visual and textual prompt learning, we set the prefix size to 16 (Zhou et al., 2022b; Jia et al., 2022). We use SGD as the optimizer, with the number of training epochs selected from $\{100, 150, 200\}$ and the learning rate selected from $\{0.001, 0.002, 0.005\}$. We adopt 16 labeled samples per class (*i.e.,* 16-shot) as the calibration set. When the base model is prompt-tuning CLIP, we split the original 16-shot calibration set into two parts: one is used for learning the prompts, and the other is reserved for training our post-hoc calibration network. Table 4 describes the detailed settings and architecture for the proposed lightweight meta network based on pretrained CLIP.

## D ADDITIONAL EXPERIMENTS

### D.1 ABLATION STUDY

The key component of the proposed method is the surrogate loss $\mathcal{L}_{SUR}$, which consists of two parts: (i) a confidence regularization term that encourages the predictions to align with the target reliability curve (*i.e.,* $\Phi(s)$), and (ii) a full-probability constraint (FPC) that complements the confidence regularization by regularizing the entire probability distribution, preventing the training instability cause by single probability (*i.e.,* confidence) were optimized. Therefore, we do not report a separate ablation with only the confidence regularization term, as it is intended to be used together with FPC. To verify the effectiveness of these components, we investigate the performance of the following variants: the original CLIP, CLIP with FPC, and CLIP with the full proposed method. The results are reported in Table 5.

According to Table 5, the complete objective function achieves the best performance. Notably, employing only partial components (*i.e.,* +FPC) also leads to improvements. Taken together, these findings verify the effectiveness of all proposed modules.

Table 5: Results of the ablation study on six datasets, measured by AUROC, FPR90-S, and FPR90-E. ↑ denotes that higher values are better, while ↓ denotes that lower values are better.

| Methods | DTD | | | Flowers102 | | | EuroSAT | | |
|---|---|---|---|---|---|---|---|---|---|
| | AUROC↑ | FPR90-S↓ | FPR90-E↓ | AUROC↑ | FPR90-S↓ | FPR90-E↓ | AUROC↑ | FPR90-S↓ | FPR90-E↓ |
| Zero-shot CLIP | 0.762 | 0.669 | 0.572 | 0.864 | 0.435 | 0.354 | 0.650 | 0.782 | 0.742 |
| +FPC | 0.795 | 0.648 | 0.471 | 0.878 | 0.388 | 0.325 | 0.778 | 0.704 | 0.471 |
| +ALL | **0.802** | **0.636** | **0.457** | **0.886** | **0.378** | **0.305** | **0.788** | **0.665** | **0.468** |

| Methods | RESICS45 | | | MNIST | | | CUB | | |
|---|---|---|---|---|---|---|---|---|---|
| | AUROC↑ | FPR90-S↓ | FPR90-E↓ | AUROC↑ | FPR90-S↓ | FPR90-E↓ | AUROC↑ | FPR90-S↓ | FPR90-E↓ |
| Zero-shot CLIP | 0.779 | 0.636 | 0.508 | 0.813 | 0.511 | 0.482 | 0.807 | 0.839 | 0.758 |
| +FPC | 0.794 | 0.615 | 0.482 | 0.908 | 0.228 | 0.235 | 0.808 | 0.613 | 0.440 |
| +ALL | **0.808** | **0.597** | **0.445** | **0.915** | **0.200** | **0.205** | **0.812** | **0.602** | **0.438** |

## D.2 PARAMETER ANALYSIS

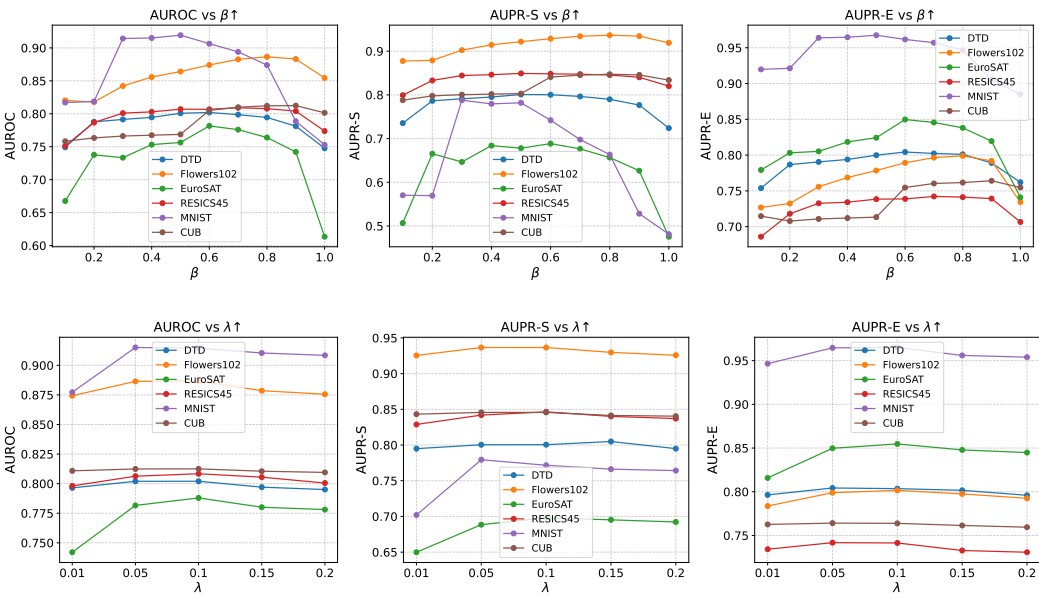

Figure 7: Line plots of model performance across six datasets (*i.e.,* DTD, Flowers102, EuroSAT, RESICS45, MNIST, and CUB) under varying hyperparameters. The first row shows the effect of $\beta$, and the second row shows the effect of $\lambda$. From left to right, each subfigure corresponds to one evaluation metric: AUROC, AUPR-S, and AUPR-E. The curves illustrate how performance changes as the hyperparameter values increase.

In the proposed method, we employ the non-negative parameters(*i.e.,* $\beta$ and $\lambda$). For $\beta$, it is used to achieve a trade-off between the two terms of the objective function (*i.e.,* $\mathcal{L}_{SUR}$). For $\lambda$, which is used to control the smoothness of the reliability curve. To investigate the impact of $\beta$ and $\lambda$ with different settings, we conduct experiments on all six datasets by varying the value of $\beta$ in the range of [0.1, 1.0] and the value of $\lambda$ in the range of [0.01, 0.2]. The results are shown in Figure 7.

From Figure 7, we make the following observations. First, for the hyperparameter $\beta$, the proposed method achieves peak performance around $\beta = 0.6$. Performance deteriorates when $\beta$ is too large or too small, as the method then fails to balance focus between correctly and incorrectly predicted samples. Second, for the hyperparameter $\lambda$, the method consistently performs well when $\lambda$ is set appropriately (e.g., [0.05, 0.1]). Similarly to $\beta$, extreme values of $\lambda$ lead to inferior performance. This is because $\lambda$ controls the smoothness of $\Psi(s)$: if $\Psi(s)$ is too sharp, the overly aggressive probability updates can cause unstable optimization; if it is too smooth, the model loses the ability to effectively adjust predictions. In practice, we find that the parameter $\lambda$ is highly stable across different datasets.

As summarized in Table 4, five out of six datasets (*i.e.,* spanning textures, fine-grained objects, satellite imagery, handwriting, and birds) select the same value $\lambda = 0.05$. This consistency indicates that $\lambda$ does not require dataset-specific tuning. Therefore, we recommend $\lambda = 0.05$ as a reliable default, which already achieves near-optimal performance in our experiments, with adjustments needed only in rare cases.

## D.3 ANALYSIS OF THE LEARNED TEMPERATURE COEFFICIENT

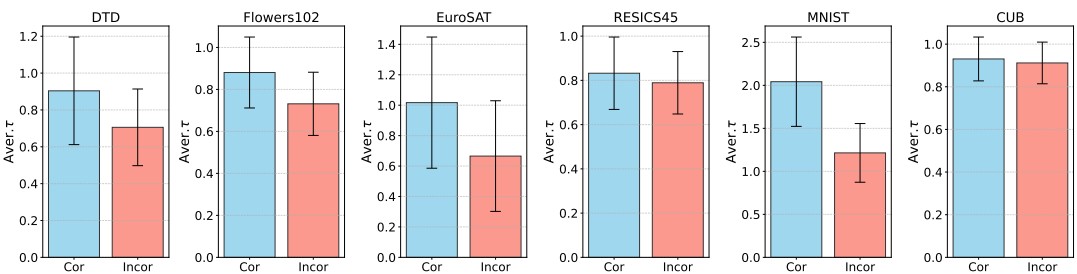

Figure 8: Comparison of learned $\tau$ values for correct versus incorrect predictions across six datasets. Each bar represents the average $\tau$ for the corresponding sample type. The blue bars represent the correctly predicted samples, and the red bars represent the incorrectly predicted samples.

To verify that the learned instance-wise $\tau$ indeed increases the confidence of correctly predicted samples compared with incorrectly predicted ones. Since the temperature coefficient $\tau$ is multiplied by the logits, a larger value of $\tau$ results in higher confidence. We compare the average learned $\tau$ values between correct and incorrect predictions on all six datasets. The results are visualized in Figure 8. In Figure 8, we can observe that the average value of the learned $\tau$ for correct predictions is larger than that for incorrect predictions, with a larger gap on most datasets. For example, on MNIST, the gap is about 68.5% of the incorrect predictions. This demonstrates that the proposed method is able to capture the intrinsic differences between correctly and incorrectly classified samples, thereby enabling their discrimination.

## D.4 RESULTS ON SIGLIP BACKBONE

To further examine whether LMN generalizes beyond the CLIP-B/32, we additionally evaluate the method using CLIP-L/14 and SigLIP-B/16 Zhai et al. (2023) as the backbones. SigLIP represents a modern vision–language architecture trained with a different alignment objective and embedding structure, offering a stronger testbed for assessing the backbone-agnostic nature of LMN. Because most existing calibration and MisD-oriented baselines do not provide publicly available or reproducible implementations for SigLIP, SCT is the only method that can be feasibly adapted to this backbone, and we therefore adopt it as the comparison baseline in this setting. The results are shown in Table 6 and Table 7.

From Table 6 and Table 7, we can observe that, across all datasets and all backbones, LMN continues to deliver consistent improvements over SCT under both AUROC and FPR90E, demonstrating that its effectiveness is not restricted to the CLIP-B/32 backbone and extends naturally to more advanced VLM architectures. These results confirm that LMN's surrogate objective and meta-network design retain their advantages even when the underlying visual encoder and multimodal representation mechanism differ substantially from those of CLIP-B/32.

## D.5 CALIBRATION DATA SENSITIVITY ANALYSIS

To evaluate whether the effectiveness of LMN depends on the specific 16-shot calibration configuration used in the main experiments, we additionally conduct a sensitivity study by varying the number of calibration samples per class. Specifically, we consider 4-shot, 8-shot, 16-shot, 32-shot, and 64-shot settings on datasets with sufficiently large training sets, including DTD, EuroSAT, RESICS45, and MNIST. For Flowers102 and CUB, the available training samples are too limited to support

Table 6: MisD Performance comparison under CLIP-L/14. For each dataset, AUROC ($\uparrow$) and FPR90E ($\downarrow$) are reported.

| Method | DTD | | Flowers102 | | EuroSAT | |
|---|---|---|---|---|---|---|
| | AUROC $\uparrow$ | FPR90-E $\downarrow$ | AUROC $\uparrow$ | FPR90-E $\downarrow$ | AUROC $\uparrow$ | FPR90-E $\downarrow$ |
| CLIP-L/14 | 0.789 | 0.521 | 0.887 | 0.289 | 0.730 | 0.605 |
| SCT | 0.808 | 0.433 | 0.903 | 0.203 | 0.787 | 0.497 |
| LMN | **0.843** | **0.373** | **0.942** | **0.132** | **0.862** | **0.351** |
| Method | RESICS45 | | MNIST | | CUB | |
| | AUROC $\uparrow$ | FPR90-E $\downarrow$ | AUROC $\uparrow$ | FPR90-E $\downarrow$ | AUROC $\uparrow$ | FPR90-E $\downarrow$ |
| CLIP-L/14 | 0.798 | 0.469 | 0.910 | 0.211 | 0.809 | 0.465 |
| SCT | 0.821 | 0.382 | 0.942 | 0.154 | 0.821 | 0.396 |
| LMN | **0.864** | **0.329** | **0.958** | **0.110** | **0.840** | **0.368** |

Table 7: MisD Performance comparison under SigLIP-B/16. For each dataset, AUROC ($\uparrow$) and FPR90E ($\downarrow$) are reported.

| Method | DTD | | Flowers102 | | EuroSAT | |
|---|---|---|---|---|---|---|
| | AUROC $\uparrow$ | FPR90-E $\downarrow$ | AUROC $\uparrow$ | FPR90-E $\downarrow$ | AUROC $\uparrow$ | FPR90-E $\downarrow$ |
| SigLIP | 0.783 | 0.376 | 0.881 | 0.337 | 0.631 | 0.746 |
| SCT | 0.804 | 0.334 | 0.898 | 0.271 | 0.641 | 0.744 |
| LMN | **0.843** | **0.292** | **0.940** | **0.107** | **0.674** | **0.713** |
| Method | RESICS45 | | MNIST | | CUB | |
| | AUROC $\uparrow$ | FPR90-E $\downarrow$ | AUROC $\uparrow$ | FPR90-E $\downarrow$ | AUROC $\uparrow$ | FPR90-E $\downarrow$ |
| SigLIP | 0.801 | 0.413 | 0.914 | 0.154 | 0.907 | 0.218 |
| SCT | 0.814 | 0.376 | 0.956 | 0.093 | 0.909 | 0.220 |
| LMN | **0.856** | **0.243** | **0.978** | **0.064** | **0.935** | **0.174** |

larger-shot configurations beyond 16-shot, and therefore, we only report results up to 16-shot for these two datasets. The results are reported in Table 8 and Table 9.

From the Tables, we can observe that across all datasets with adequate training data, both AUROC and FPR90E exhibit stable and progressively improving trends as the number of calibration samples increases. This behavior is consistent with the design of LMN: as a lightweight meta-network, LMN naturally benefits from richer calibration signals, while its overall performance remains robust even under very low-shot conditions, such as 4-shot and 8-shot. These observations indicate that LMN's effectiveness does not depend on the specific 16-shot setup used in the main experiment.

### D.6 ANALYZING THE COMPUTATIONAL EFFICIENCY OF LMN

To quantify the computational overhead of the proposed Lightweight Meta Network (LMN), we report the number of parameters, training time, and inference time on all six datasets in the Table 10.

First, for the number of parameters, across all datasets, LMN contains only 17K–20K parameters, which is less than 0.02% of the 151M parameters in CLIP ViT-B/32. This demonstrates that LMN is extremely lightweight relative to the backbone model.

Second, for the training and test time, the results show that LMN is highly efficient in practice. All reported timings already include the one-time CLIP forward pass required to compute image and text embeddings, making the measurements fully reflect the real end-to-end cost. Training on all datasets completes within one minute, with most datasets requiring only a few seconds (e.g., 7.05s on EuroSAT and 13.31s on DTD). This confirms that LMN introduces negligible computational burden during optimization. Inference is similarly lightweight. Even on RESICS45, which contains over 20,000 test samples, LMN requires only 339 seconds ( 5 minutes) to process the entire test set.

Table 8: AUROC (↑) under different numbers of calibration shots. '-' denotes that the result cannot be obtained.

| Dataset | CLIP | 4-shot | 8-shot | 16-shot | 32-shot | 64-shot |
|---|---|---|---|---|---|---|
| DTD | 0.762 | 0.787 | 0.779 | 0.792 | 0.804 | 0.807 |
| Flowers102 | 0.864 | 0.873 | 0.884 | 0.886 | – | – |
| EuroSAT | 0.650 | 0.732 | 0.765 | 0.788 | 0.792 | 0.797 |
| RESICS45 | 0.779 | 0.793 | 0.804 | 0.808 | 0.810 | 0.815 |
| MNIST | 0.813 | 0.883 | 0.901 | 0.915 | 0.936 | 0.941 |
| CUB | 0.807 | 0.810 | 0.810 | 0.812 | – | – |

Table 9: FPR90E (↓) under different numbers of calibration shots. CLIP denotes the zero-shot baseline.

| Dataset | CLIP | 4-shot | 8-shot | 16-shot | 32-shot | 64-shot |
|---|---|---|---|---|---|---|
| DTD | 0.572 | 0.512 | 0.474 | 0.504 | 0.438 | 0.431 |
| Flowers102 | 0.354 | 0.329 | 0.317 | 0.305 | – | – |
| EuroSAT | 0.742 | 0.619 | 0.522 | 0.468 | 0.450 | 0.439 |
| RESICS45 | 0.508 | 0.477 | 0.462 | 0.445 | 0.446 | 0.421 |
| MNIST | 0.482 | 0.301 | 0.264 | 0.205 | 0.160 | 0.142 |
| CUB | 0.554 | 0.549 | 0.538 | 0.532 | – | – |

For smaller datasets, inference finishes within tens of seconds (e.g., 15.77s on DTD and 53.83s on EuroSAT). These results validate that LMN remains practical and efficient even when applied to large-scale test sets.

Overall, the empirical measurements demonstrate that LMN maintains its lightweight property in both training and inference, making it suitable for real-world and large-scale deployments.

## D.7 EVALUATION UNDER DISTRIBUTION SHIFT

To further assess the robustness of the proposed LMN under distribution shift, we conduct an additional evaluation using the ImageNet-Val to train the LMN and evaluating on two distribution-shifted benchmarks (*i.e.,* ImageNet-A and ImageNet-Sketch). Specifically, ImageNet-Sketch, which introduces substantial style and texture shift, and ImageNet-A, which contains adversarially curated natural images designed to induce model failures. We report both misclassification-detection metrics (AUROC and FPR90) and calibration metrics (Brier score) to obtain a comprehensive view of the model's behavior under shift. The results are provided in Table 11.

From the Table 11, we can observe that LMN consistently improves AUROC and reduces FPR90-E across both distribution-shifted datasets, indicating stronger discrimination between correct and incorrect predictions in the presence of distribution shifts. Meanwhile, LMN maintains competitive calibration performance. These findings confirm that LMN remains effective even when the input distribution deviates significantly from the calibration set, further demonstrating the practicality of the proposed approach.

Table 10: Training/inference time and number of LMN parameters across six datasets. LMN is highly lightweight, with only 17K–20K parameters and second-level training time on all datasets.

| Time (s) | DTD | Flowers102 | EuroSAT | RESICS45 | MNIST | CUB |
|---|---|---|---|---|---|---|
| Train | 13.31 | 29.63 | 7.05 | 13.14 | 6.81 | 48.35 |
| Test | 15.77 | 113.58 | 53.83 | 339.01 | 96.07 | 95.48 |
| **Num. Parameters** | **DTD** | **Flowers102** | **EuroSAT** | **RESICS45** | **MNIST** | **CUB** |
| LMN | 17.2K | 18.8K | 17.3K | 18.1K | 17.3K | 20.3K |

Table 11: Performance under distribution shift on ImageNet-A and ImageNet-Sketch.

| Method | ImageNet-A | | | ImageNet-Sketch | | |
|---|---|---|---|---|---|---|
| | AUROC↑ | FPR90-E↓ | Brier↓ | AUROC↑ | FPR90-E↓ | Brier↓ |
| Zero-shot CLIP | 0.657 | 0.740 | 0.814 | 0.805 | 0.495 | 0.732 |
| LMN | **0.682** | **0.716** | **0.796** | **0.813** | **0.484** | **0.721** |

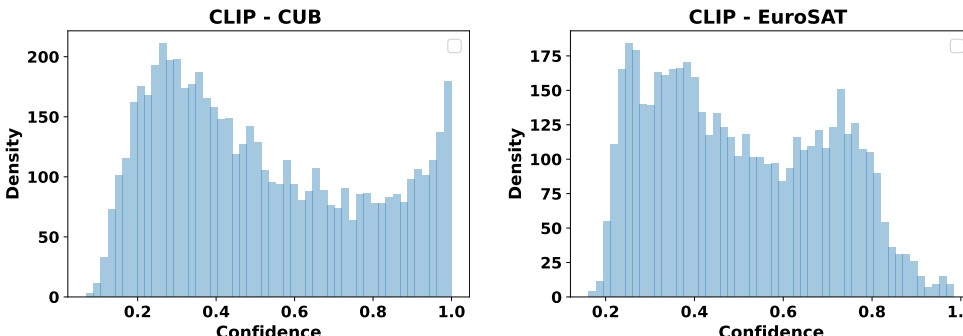

Figure 9: Histogram of test-sample confidence levels for CLIP on the CUB (left) and EuroSAT (right) datasets.

### D.8 ADDITIONAL EMPIRICALLY SUPPORT

We empirically examine the confidence distribution on two benchmark datasets: CUB and EuroSAT, respectively. For the test set, we record the confidence assigned by the pretrained CLIP model to each sample. We then visualize the distribution using histograms to visualize the density of each confidence level. The results are shown in Figure 9.

Empirically, as shown in Figure 9, CLIP confidence values on test samples are widely spread across the $[0, 1]$ interval, with many samples falling in the middle range. As a result, the conditional expectations in $[r, 1]$ for correct prediction detection and in $[0, r]$ for incorrect prediction detection are far below 1, confirming that strict confidence calibration alone cannot achieve high MisD performance.

## E LLM USAGE STATEMENT

In this work, a large language model (LLM) was used to polish the writing. The LLM assisted in improving clarity and grammar, but all scientific content, interpretations, and conclusions were generated solely by the authors.

