# OpenReview forum: "Revisiting Confidence Calibration for Misclassification Detection in VLMs"
_ICLR.cc/2026/Conference — ICLR 2026 Poster_

### Official Review · Reviewer_x3hT · 2025-10-25

**Soundness:** 3
**Presentation:** 3
**Contribution:** 3
**Rating:** 6
**Confidence:** 4

**Summary:**

This paper addresses misclassification detection (MisD) in vision-language models like CLIP. The author first shows that the objective of confidence calibration is misaligned with MisD. Then, the authors theoretically analyze the limitations of standard calibration objectives in differentiating correct from incorrect predictions. To solve the issue, they design a differentiable surrogate loss to increase separation between correct and incorrect predictions. Extensive experiments across six datasets demonstrate that LMN can improve MisD performance compared to recent calibration or OOD detection methods.

**Strengths:**

1.	The motivation is well-structured. The paper first clearly distinguishes the goal of confidence calibration from misclassification detection, and show that a well-calibrated model can be worse for MisD than a poorly-calibrated model.
2.	The theoretical analysis is clear. The author links the area under/above the curve in the reliability diagram to the precision of detecting correct/incorrect predictions, and show that even a perfectly calibrated mode has limited MisD precision, bounded by the conditional expectation of confidence.
3.	The experimental results are technologically sound and generally support the author’s claim. The author show that previous calibration can not bring significant improvement in MisD and the proposed method LMN shows consistent improvements across diverse datasets.

**Weaknesses:**

1.	Lack of quantitative analysis for the motivation. The paper provides a strong conceptual illustration of the failure of calibration in Figure 1. However, it lacks a quantitative experiment to directly quantify the MisD performance after calibration (e.g., TS[1] or ProximityBias [2]).
2.	The connection between the proposed method and VLM is weak. The proposed LMN relies only on output logits. Consequently, the method appears applicable to any classifier rather than being inherently tied to VLMs. The author carefully discusses the role of the proposed LMN in VLM.
3.	Missing comparison with other failure detection methods like [3]. The experiments only compare with general calibration or OOD detection methods for VLM.
4.	Undefined loss notation. $L_{LDA}$ in Appendix E.2 is not defined anywhere in the paper. It looks like a typo for $L_{SUR}$.

[1] On calibration of modern neural networks. ICML, 2017.

[2] Proximity-informed calibration for deep neural networks. NeurlPS, 2023.

[3] Rethinking confidence calibration for failure prediction. ECCV, 2022.

**Questions:**

Questions:
1.	How does LMN affect overall calibration performance?  It is unclear whether LMN compromises or preserves global calibration alignment.
2.	How does DOR perform in the open-vocabulary setting? Table 2 presents open-vocabulary results for the proposed method. However, it does not compare with open-vocabulary calibration methods like DOR [4].

[4] Understanding and mitigating miscalibration in prompt tuning for vision-language models.

---

> ### Author Response · Authors · 2025-11-20
> **Response to Reviewer x3hT (part 1)**
>
> We thank the reviewer for the positive assessment and the constructive suggestions. Our point-by-point responses are provided below.
>
> >**W1**: Lack of quantitative analysis for the motivation. The paper provides a strong conceptual illustration of the failure of calibration in Figure 1. However, it lacks a quantitative experiment to directly quantify the MisD performance after calibration (e.g., TS[1] or ProximityBias [2]).
>
> **A1**: Thank you for the comment. Table 1 already includes two recent calibration-based approaches, FeatureClipping and DOR, and their MisD performance is reported directly. These results show that existing calibration methods provide only marginal gains in MisD.
> Regarding Temperature Scaling (TS), it applies the same scaling factor to the logits of all samples. Since different samples have different logit distributions, this uniform rescaling does not meaningfully change their confidence ordering. As MisD metrics rely on ranking correct predictions above incorrect ones, TS is not expected to improve MisD performance.
> For completeness, we also evaluate TS and ProximityBias. The results indicate that neither method yields noticeable improvements in MisD performance.
>
> | AUROC↑        | DTD   | FLowers102 | EuroSAT | RESICS45 | MNIST | CUB   |
> |-------------|-------|------------|---------|----------|-------|-------|
> | CLIP          | 0.762 | 0.864      | 0.65    | 0.779    | 0.813 | 0.807 |
> | TS            | 0.760 | 0.856      | 0.644    | 0.777    | 0.810 | 0.804 |
> | ProximityBias | 0.764 | 0.853      | 0.651   | 0.782    | 0.823 | 0.805 |
>
>
> | FPR90E↓       | DTD   | FLowers102 | EuroSAT | RESICS45 | MNIST | CUB   |
> |--------------|-------|------------|---------|----------|-------|-------|
> | CLIP          | 0.572 | 0.354      | 0.742   | 0.508    | 0.482 | 0.442 |
> | TS            | 0.564 | 0.363      | 0.735   | 0.520    | 0.488 | 0.458 |
> | ProximityBias | 0.563 | 0.371      | 0.738   | 0.498    | 0.452 | 0.450 |
>
>
> >**W2**: The connection between the proposed method and VLM is weak. The proposed LMN relies only on output logits. Consequently, the method appears applicable to any classifier rather than being inherently tied to VLMs. The author carefully discusses the role of the proposed LMN in VLM.
>
> **A2**: We thank the reviewer for this insightful observation. Our method is conceptually model-agnostic, since the surrogate objective operates on predicted confidence scores and could in principle be applied to other classifiers. However, in our implementation we explicitly make use of multimodal information unique to VLMs, including the image embedding, the predicted text embedding, and the way logits reflect image–text semantic interactions. These signals capture alignment properties specific to VLMs and are not available in unimodal models. We also provide an analysis of the contribution of each modality to LMN in Appendix C.2, which further illustrates the role of VLM-specific features in our design.
>
> Furthermore, VLMs are pretrained on large-scale image–text corpora and possess strong zero-shot and open-vocabulary generalization capabilities, which make them the closest to real-world deployment among current vision models. Misclassification detection in such settings is substantially more challenging and practically relevant. For this reason, coupling our method with VLMs is meaningful. In many application domains, practitioners directly adopt pretrained VLMs without the need for repeated training, and reliable misclassification detection on these pretrained models is therefore both necessary and impactful.
>
>
> >**W3**: Missing comparison with other failure detection methods like [3]. The experiments only compare with general calibration or OOD detection methods for VLM.
>
> **A3**: Thank you for raising this point. The method in [3] requires searching for flat minima during model training, which involves updating the full set of model parameters. This procedure is not suitable for pretrained large model, such as CLIP.
>
> To provide a fair and relevant comparison, we include post-hoc failure-detection method that can be applied directly to pretrained VLMs. We have added ViLU (2025,ICCV)in the revised manuscript (Table 1, highlighted in blue), and the results show that our method still achieves a clear advantage over this baseline.
>
>
> >**W4**: Undefined loss notation. $L _{LDA}$ in Appendix E.2 is not defined anywhere in the paper. It looks like a typo for $L _{SUR}$.
>
> A4: Thank you for catching this typo. The term $L _{LDA}$ in Appendix E.2 was unintended and should be $L _{SUR}$. We have corrected the notation in the revised manuscript.

---

> ### Author Response · Authors · 2025-11-20
> **Response to Reviewer x3hT (part 2)**
>
> >**Question 1**: How does LMN affect overall calibration performance? It is unclear whether LMN compromises or preserves global calibration alignment.
>
> **A1** : We thank the reviewer for this question. The LMN is designed to optimize a MisD-oriented calibration objective rather than the classical diagonal calibration objective that aligns confidence with empirical accuracy. As shown in our theoretical analysis (Section 3.2), these two objectives exhibit an inherent tension: the diagonal reliability curve (i.e., the perfect-calibration target) constrains the confidence distribution in a way that limits the achievable MisD performance. This observation motivates us to design a different reliability-curve target that is explicitly aligned with MisD. As a consequence, the two objectives are difficult to optimize simultaneously, and explicitly identifying this distinction and developing a MisD-oriented reliability target forms one of the key contributions of this paper.
>
> To address your concern, we additionally evaluated the calibration performance of LMN and observed that LMN can indeed improve the overall calibration performance (measured by Birer) when starting from the zero-shot CLIP. However, because LMN is not designed to align predictions with the diagonal (acc = conf), it does not aim to achieve state-of-the-art calibration performance. This behavior is expected and consistent with the theoretical trade-off between calibration and MisD.
>
> | Birer↓ | DTD | FLowers102 | EuroSAT | RESICS45| MNIST   | CUB       |
> |-------|----|-------|-------|----|-----------|-----------|
> | CLIP  | 0.723 | 0.496  | 0.815   | 0.627   | 0.855  | 0.618 |
> | LMN(ours)| **0.699** | **0.475**  | **0.728** | **0.586** | **0.815** | **0.608** |
>
>
> >**Question 2**. How does DOR perform in the open-vocabulary setting? Table 2 presents open-vocabulary results for the proposed method. However, it does not compare with open-vocabulary calibration methods like DOR [4].
>
> **A2**: Thank you for pointing this out. We have now included the open-vocabulary results of DOR in the revised manuscript. The updated comparisons are provided in Table 2 (highlighted in blue), where we evaluate DOR under the same open-vocabulary protocol as LMN and zero-shot CLIP. The results show that although DOR provides modest improvements over zero-shot CLIP in some cases, LMN consistently achieves stronger misclassification-detection performance across most datasets and metrics.
>
> | AUROC↑         | DTD       | FLowers102 | EuroSAT   | RESICS45  | MNIST     | CUB       |
> |----------------|-----------|------------|-----------|-----------|-----------|-----------|
> | Zero-shot CLIP | 0.760     | 0.853      | 0.608     | 0.780     | 0.854     | 0.789     |
> | DOR            | 0.758     | 0.847      | 0.605     | 0.764     | 0.842     | 0.779     |
> | LMN(ours)      | **0.770** | **0.858**  | **0.614** | **0.782** | **0.859** | **0.790** |
>
> | FPR90E↓        | DTD       | FLowers102 | EuroSAT   | RESICS45  | MNIST     | CUB       |
> |----------------|-----------|------------|-----------|-----------|-----------|-----------|
> | Zero-shot CLIP | 0.604     | 0.397      | 0.882     | 0.536     | 0.322     | 0.490     |
> | DOR            | 0.612     | 0.394      | 0.883     | 0.543     | 0.331     | 0.512     |
> | LMN(ours)      | **0.598** | **0.387**  | **0.875** | **0.527** | **0.301** | **0.472** |

---

> > ### Comment · Reviewer_x3hT · 2025-11-22
> >
> > Thanks for the rebuttal. In short, most of the concerns have been solved. I will keep my positive score. One potential concern is that the connection between LMN and CLIP is still weak.

---

> > > ### Author Response · Authors · 2025-11-22
> > > **Thanks for your reply!**
> > >
> > > Thank you for the follow-up and for maintaining your positive evaluation. Regarding the remaining concern, we would like to briefly note that LMN is indeed closely connected to VLMs in practice. It leverages multimodal information such as the image embedding, the predicted text embedding, and the image–text interaction encoded in VLM logits, which is not accessible in unimodal classifiers. As discussed in Appendix C.2 of the revised manuscript, these three types of signals contribute complementary and non-redundant information: logits characterize the confidence geometry, image embeddings reflect sample-level visual difficulty, and predicted text embeddings reveal semantic confusion patterns across categories. Together, they provide LMN with VLM-specific cues that meaningfully influence the estimation of temperature factors and are essential to its overall behavior.

---

### Official Review · Reviewer_Mme6 · 2025-10-30

**Soundness:** 2
**Presentation:** 1
**Contribution:** 3
**Rating:** 2
**Confidence:** 4

**Summary:**

This paper investigates the connection between confidence calibration and misclassification detection (MisD), revealing that standard calibration methods limit misclassification detection (MisD) performance. The authors propose a new reliability curve designed to improve the precision of detecting both correct and incorrect predictions by better separating them. To address the non-differentiability of this new curve, a surrogate loss is introduced, and a lightweight post-hoc framework employing a meta network for sample-specific temperature scaling is developed to preserve the predictive power of Vision-Language Models (VLMs). Both theoretical analysis and experiments demonstrate that this approach consistently enhances MisD performance without sacrificing model accuracy.

**Strengths:**

The paper addresses an important and yet complex challenge in any classification network. The authors focus on VLMs and misclassification performance. The paper nicely combines a mathematical founded treatment and empirical experiments to validate their findings. This provides a robust backdrop for the presented scientific approach which can be followed with some investment for any reader. Moreover, the topic it addresses is of interest to practical applications of VLMs.

Further:
- Publicly available, reproducible code.
- Comprehensive theoretical analysis supported by empirical results.
- Clear originality: reframing the relationship between calibration targets and separability of correct vs. incorrect predictions.
- Consistent improvements demonstrated across datasets and settings.
- Post-hoc design maintains the zero-shot generalization ability of VLMs while enhancing reliability.

**Weaknesses:**

The paper suffers from very imprecise language while presupposing a lot of knowledge from the reader. Moreover, does the style of the paper recall a blog post at many points. The mathematical treatment of the subject eliviates this to some extent as it provides a rigorous means of communication. However, especially towards the end of the report where the empirical findings are meant to support the theoretical treatment, the paper lacks scrutiny (missing uncertainty assessment of the presented results in Table 1 and 2 as well as no uncertainty estimates in figure 3 and 4). This renders the presentation of evidence weak as many of the presented performance metrics only differ by singular percentages to the competition.

Further:
- The originality of the surrogate loss formulation is somewhat unclear. The core novelty may primarily lie in the use of a sigmoid-shaped target curve, and it would help to clarify how this extends beyond a standard activation-based reformulation.
- The distinction between the *original calibration method* (theoretical objective) and the *post-hoc meta-network implementation* is often ambiguous, especially in the Experiments and Evaluation sections.
- The choice of (r=0.5) in defining the new target calibration curve (Figure 2) should be theoretically or empirically motivated.
- Some notations (especially those connecting the reliability curve to the surrogate penalty) are heavy, making it hard for readers to see how the calibration diagonal relates to the standard loss.
- The paper does not explore implications for related uncertainty-based tasks (e.g., OOD detection) or visualize the distribution of confidence scores.
- Key summary metrics such as **ECE** and **Accuracy** are missing from the results tables, though both are important for assessing calibration and interpreting AUROC.
- Some conclusions slightly overstate empirical findings (e.g., "but also empirically corroborate the theoretical analysis, validating the advantages of the normalized sigmoid reliability curve in MisD." - based on the calibration curve in Figure 4, which is not defined enough for me to agree that this corroborates the theoretial analysis).
- Information about dataset size, accuracy and diversity are missing, making it hard to understand how generalizable these findings are

**Questions:**

- line 32: "easily deployed" please remove such heuristic andcolloguial statements
- line 35: "the reliability of VLMs is often overlooked" this is true for many deep learning models, please remove
- line 36: "has emerged as a key research direction for building reliable VLMs." this is true for many deep learning models, please remove
- line 44-49: "To address the former, ..." such relative statements are hard to follow. Please reformulate these lines for better readability and clarity
- page 2, Fig 1:
    + many coloers in the plot, this hinders readability
    + remove superfluous backgrounds (grey)
    + remove emojis, the reader should make interpretations not the figure
    + axis title of fig 1b, should be confidence*s*
    + the threshold in fig 1b is very small
    + remove superflous boxes with rounded corners (this distracts the very complicated figure)
    + caption: remove the literal reference to concrete confidence scores, these should go into a table if important
    + line 67-71: this is one long sentence, please reformulate
- line 75-80: "To remedy" is one long sentence, please split it up for readibility
- line 87-96: the bullet points repeat the text above, please keep only one to avoid confusion and leave space for other improvements
- line 106: `sim` is not explained anywhere
- line 108: Start sentence, "Where the "{class}" is filled.
- line 148: "diagonal (y=x)" which should be "diagonal (acc = s)", possibly repeat that you mean accuracy versus confidence
- page 3: lemma 3.1 appears to be independent of `r` - if I confused this, please consider reformulating as the math suggests as much
- page 4, equation 5: $\lambda$ should be included in the function definion left of $\Psi$ to indicate a dependency
- page 5, line 254/255: "and is more flexible than perfect calibration" this appears to be unwarranted claim, please remove or reformulate
- page 6, line 283-292: very hard to follow paragraph. Please consider reformulating it to highlight central ideas.
- Table 1: no error bars or statistical treatment which elucidates competitiveness, not discussion of variability of metric results in the text too
- line 360: "we adopt widely used metrics" this is jargon, please remove.
- figure 3: only small differences visible, an uncertainty treatment would make this plot stronger
- line 413: very general claim about universality of your approach, as the data can be consdired weak, please remove
- table 2: no error bars or uncertainty treatment

---

> ### Author Response · Authors · 2025-11-20
> **Response to Reviewer Mme6 (part 1)**
>
> Thank you for your careful review and constructive comments. We provide responses to the raised concerns below.
>
> >General Response on Writing Clarity and Presentation
>
> We sincerely thank the reviewer for the detailed comments regarding writing clarity, notation, figure design, and empirical presentation. We appreciate the reviewer’s effort in providing such comprehensive feedback, and we carefully revised the manuscript to address these issues and improve the overall readability and precision of the paper.
>
> In the revised version, we have refined the exposition, simplified notational conventions, and restructured several long or unclear sentences. We also made targeted improvements to figure clarity, such as removing unnecessary visual elements and enhancing readability, while keeping the original content unchanged. In addition, we revised statements that might appear ambiguous or overly strong, ensuring a more precise and rigorous tone throughout the manuscript. Although a few stylistic suggestions did not fully align with our intended presentation style, we have incorporated the majority of the reviewer’s recommendations to substantially enhance the clarity and consistency of the paper.
>
> All corresponding changes have been integrated into the revised manuscript and are highlighted in blue for ease of reference.
>
> >**W1**: The originality of the surrogate loss formulation is somewhat unclear. The core novelty may primarily lie in the use of a sigmoid-shaped target curve, and it would help to clarify how this extends beyond a standard activation-based reformulation.
>
> **A1**: The proposed surrogate loss is not obtained by simply applying an activation function to the logits. Activation-based methods operate locally, transforming each logit pointwise without encoding any global constraint on the confidence–accuracy relationship.
>
> In contrast, our formulation penalizes deviations from the sigmoid-shaped target reliability curve, which represents the desired global structure of how accuracy should vary with confidence. For example, when a misclassified sample receives high confidence, its accuracy (0%) lies far below the expected value $ \Psi(s)$ on the sigmoid reliability curve at that confidence level. The loss term $ \Psi(s) - 0$ therefore penalizes the sample proportionally to this gap, reducing upward distortion in the high-confidence region of the reliability curve. Similarly, correctly classified samples with low confidence are penalized according to $1 -  \Psi(s)$, encouraging them to move upward toward the curve’s expected accuracy in the low-confidence region.
>
> Thus, the surrogate loss directly optimizes curve-level alignment by weighting each sample according to its deviation from the sigmoid reliability curve. This global curve-fitting objective cannot be captured by activation-based pointwise transformations, which do not impose any structural constraint across confidence levels.
>
> >**W2**: The distinction between the original calibration method (theoretical objective) and the post-hoc meta-network implementation is often ambiguous, especially in the Experiments and Evaluation sections.
>
>
> **A2**: We thank the reviewer for highlighting this. Our method indeed contains two distinct components:
> (1) the theoretical calibration objective, which defines the normalized sigmoid reliability curve $ \Psi(s)$ and its desirable properties; and
> (2) the post-hoc implementation, which learns sample-specific temperature factors to approximate this global objective.
>
> The theoretical reliability curve cannot be optimized directly, because it depends on the population-level accuracy at each confidence level—a quantity that can only be estimated after aggregating many samples. This is a common challenge in calibration, and existing approaches (e.g., histogram binning, isotonic regression, proxy losses) also rely on surrogate formulations to approximate the ideal reliability curve.
>
> In our case, we design a new surrogate loss that allows the meta-network to approximate the target sigmoid-shaped curve by learning a distinct temperature value for each sample, thereby adjusting its confidence score toward the desired curve. This forms the bridge between the theoretical objective and the practical post-hoc calibration mechanism.

---

> ### Author Response · Authors · 2025-11-20
> **Response to Reviewer Mme6 (part 2)**
>
> >**W3**: The choice of (r=0.5) in defining the new target calibration curve (Figure 2) should be theoretically or empirically motivated.
>
>
> **A3**: We thank the reviewer for raising this point. The choice of $r=0.5$ is guided by both calibration intuition and structural considerations.
>
> (1) MisD intuition and the natural decision threshold. In confidence-based discrimination, 0.5 is the standard neutral threshold separating the “uncertain” and “confident” regions.This is consistent with conventional binary calibration principles [1].
>
> [1]Confidence classifiers with guaranteed accuracy or precision. Conformal and Probabilistic Prediction with Applications.2023
>
>
> (2) Symmetry and avoiding bias toward either side of the confidence range. The midpoint r determines where the sigmoid curve changes from slow to rapid growth. Choosing $r=0.5$ ensures that the curve is symmetric with respect to the confidence domain [0,1]. A non-symmetric choice (e.g., shifting r to the left or right) would inherently bias the calibration toward either low-confidence or high-confidence regions, which is undesirable for a general-purpose MisD objective.
>
> We have clarified these motivations in Appendix C.3 of the revised manuscript.
>
>
> >**W4**: Some notations (especially those connecting the reliability curve to the surrogate penalty) are heavy, making it hard for readers to see how the calibration diagonal relates to the standard loss.
>
> **A4**: Thank you for the suggestion. We will further refine and carefully check the relevant parts.
>
>
> >**W5**: The paper does not explore implications for related uncertainty-based tasks (e.g., OOD detection) or visualize the distribution of confidence scores.
>
> **A5**: We thank the reviewer for raising this point. While our main focus is MisD, we agree that implications for uncertainty-based tasks such as OOD detection are important. Although we did not explicitly frame our open-vocabulary experiments as OOD evaluation, this setting naturally induces an OOD-like scenario: during testing, the model encounters samples from unseen categories, which aligns closely with semantic OOD detection in vision–language models. As shown in Section 4.4, our method already improves the separation between correct and incorrect predictions on unseen classes, which reflects stronger epistemic uncertainty estimation.
> To further address the your concern, we additionally include a dedicated OOD detection experiment following the protocol of [1]. We use a subset of ImageNet as the in-distribution data and evaluate OOD performance on subsets of iNaturalist and Textures. We compare LMN against SCT, a recent state-of-the-art OOD detection method for VLMs. The results, provided in following Table, show that LMN achieves higher AUROC and lower FPR90 than SCT on both datasets, indicating stronger OOD discrimination.
>
> | iNaturalist-AUROC↑ | Textures-FPR90-E↓  | Textures-AUROC↑ | Textures-FPR90-E↓ |
> |---------|------------|------------|-------------|
> | SCT | 0.959            | 0.089            | 0.891          | 0.343            |
> | LMN | 0.961             | 0.074             | 0.913          | 0.301            |
>
>
> Finally, regarding the distribution of confidence scores, we provide an analysis in Appendix E.3 where we visualize the learned temperature coefficients. These coefficients directly reflect how LMN adjusts confidence for different samples. The results show a clear pattern: LMN increases the confidence of correctly classified samples while reducing the confidence assigned to misclassified ones, consistent with its goal of improving MisD and OOD separability.
>
> [1] Mos: Towards scaling out-of-distribution detection for large semantic space.CVPR,2021.
>
>
>
> >**W6**: Key summary metrics such as ECE and Accuracy are missing from the results tables, though both are important for assessing calibration and interpreting AUROC.
>
>
> **A6**:  As our method is a post-hoc calibration approach, it adjusts only the confidence scores through sample-specific temperature factors and therefore does not change the model’s classification accuracy. For this reason, accuracy remains identical to the base VLM and was omitted from the tables.
>
> Regarding ECE, we agree that it is a widely used metric for evaluating confidence calibration. Specifically, ECE measures how well a model’s predicted confidence aligns with the empirical accuracy, and is therefore designed to assess the calibration objective rather than discrimination performance. However, as shown in our theoretical analysis in Section 3.2, calibration (and its corresponding metric ECE) does not capture the ability to distinguish correct from incorrect predictions—the key requirement of MisD. Thus, ECE cannot reflect the prediction-level separation that MisD relies on, even though it remains a standard metric for assessing calibration quality.

---

> ### Author Response · Authors · 2025-11-20
> **Response to Reviewer Mme6 (part 3)**
>
> >**W7**: Some conclusions slightly overstate empirical findings (e.g., "but also empirically corroborate the theoretical analysis, validating the advantages of the normalized sigmoid reliability curve in MisD." - based on the calibration curve in Figure 4, which is not defined enough for me to agree that this corroborates the theoretial analysis).
>
> **A7**: Thank you for pointing this out. We have revised the sentence in the manuscript to remove the overstatement.
>
>
> >**W8**: Information about dataset size, accuracy and diversity are missing, making it hard to understand how generalizable these findings are.
>
> **A8**: The dataset statistics, including the number of classes and the sizes of the training and test sets, are already provided in Appendix Table 3. Regarding accuracy, our method is a post-hoc approach and does not modify the backbone model’s parameters and predictions, so the base accuracy remains unchanged. Our focus is on improving misclassification detection rather than altering classification accuracy.
>
> > **For all questions**:
>
> We thank the reviewer for the detailed line-by-line suggestions regarding wording, clarity, notation, and figure design.We have incorporated these constructive suggestions into our revised manuscript, with all changes highlighted in blue. For a few other comments that may have arisen from ambiguities in our original text, we provide clarifications below.
>
>
> >**Question**: page 3: lemma 3.1 appears to be independent of r - if I confused this, please consider reformulating as the math suggests as much.
>
> **A**: Lemma 3.1 is indeed presented in a general form that holds for any interval [a, b] . The case involving the threshold $ r$, where correctly predicted samples fall into $[r, 1]$ and incorrectly predicted ones into $[0, r]$, is a specific instance covered by our lemma. This is achieved by setting $[a, b]$ to $[r, 1]$ or $[0, r]$ in our formulation.

---

> ### Author Response · Authors · 2025-11-26
> **Thank you for the encouraging feedback!**
>
> Thank you very much for raising your score. Your careful reading and detailed suggestions greatly helped us improve the quality of this paper. We will add the indication of variability and uncertainty (i.e., statistical error) accordingly.

---

### Official Review · Reviewer_jRJV · 2025-10-30

**Soundness:** 3
**Presentation:** 3
**Contribution:** 3
**Rating:** 6
**Confidence:** 3

**Summary:**

This paper studies misclassification detection (MisD) for VLMs and argues that standard confidence calibration can hinder MisD. Specifically, the author argues that the objective of aligning confidence with accuracy is theoretically at odds with the goal of MiSD, which requires a clear ranking between correct and incorrect predictions. To address this issue, this paper proposes a recalibration objective based on a normalized sigmoid reliability curve, which is designed to maximize the separation between correct and incorrect predictions at every confidence level. Since this objective is non-differentiable, the authors introduce a differentiable surrogate loss. To preserve the original VLM's capabilities, they implement this via a post-hoc, lightweight meta-network that predicts instance-specific temperature scaling factors. Empirical experiments on multiple datasets demonstrate the effectiveness of the proposed method.

**Strengths:**

1. The core argument that perfect calibration inherently limits MiSD performance is technically sound.
2. The empirical results demonstrate the effectiveness of the proposed method.
3. The paper is well written.

**Weaknesses:**

1. The parameter $\lambda$, which controls the sharpness of the sigmoid curve, is crucial for balancing separation strength and stability. While a parameter analysis is provided, the paper could offer more concrete guidance on how to select $\lambda$ in practice for a new dataset, rather than treating it purely as a hyperparameter to be tuned.
2. Results focus on CLIP ViT-B/32 (plus prompt variants). Including B/16, L/14, or OpenCLIP variants, and at least one non-CLIP VLM would benefit this paper.
3. The “lightweight” claim is plausible, but concrete numbers for training/inference overhead would still benefit the paper.

**Questions:**

1. In the surrogate loss (Eq. 8), the fusion function $\phi$ can be summation or multiplication. Table 4 shows it is set to "summation" for most datasets but "multiply" for RESICS45 and CUB. What is the intuition behind this choice? Is there a clear rule or characteristic of a dataset that would guide this selection?
2. In Table 1, the performance of the proposed method on Flowers102, RESICS45, and CUB seems marginal. Could you analyze the reason? And does this mean the effectiveness of the proposed method is dataset dependent?
3. The meta-network uses features from both image and text modalities. Could you provide an ablation or insight to understand the relative importance of each modality's features (image embedding, text embedding, logits) for predicting the effective temperature factor?

---

> ### Author Response · Authors · 2025-11-20
> **Response to Reviewer jRJV (part 1)**
>
> We thank the reviewer for the positive assessment and the constructive suggestions. Our point-by-point responses are provided below.
>
> >**W1**: The parameter $ \lambda$, which controls the sharpness of the sigmoid curve, is crucial for balancing separation strength and stability. While a parameter analysis is provided, the paper could offer more concrete guidance on how to select in practice for a new dataset, rather than treating it purely as a hyperparameter to be tuned.
>
> **A1**: We appreciate the reviewer’s comment. While $\lambda$ controls the sharpness of the proposed sigmoid calibration curve, our empirical results show that it is in fact highly stable across different domains. As reported in Table 4, five of the six datasets, which cover textures, fine-grained objects, satellite imagery, handwriting, and birds, choose the same value $\lambda$ = 0.05 as the best-performing setting. This strong consistency indicates that $\lambda$ does not require dataset-specific tuning and is largely robust to domain differences.
>
> In practice, $\lambda$ = 0.05 serves as a reliable default and already achieves near-optimal performance across all tested datasets. Users may only need to adjust it in rare cases.We have added this practical guideline to Appendix E.2 (highlighted in blue).
>
>
> >**W2**: Results focus on CLIP ViT-B/32 (plus prompt variants). Including B/16, L/14, or OpenCLIP variants, and at least one non-CLIP VLM would benefit this paper.
>
> **A2**: We thank the reviewer for this valuable suggestion. Our method is fully post-hoc, meaning that it does not rely on any specific backbone architecture and does not modify the image/text embeddings or the logits of the underlying VLM. Therefore, it is naturally compatible with a wide range of VLM backbones.
> To further demonstrate this generality, we have added experiments on CLIP ViT-L/14 and SigLIP, two widely used and stronger VLM backbones. Since many calibration and MisD baselines do not provide implementations for SigLIP or other non-CLIP models, we adopt SCT, which can be easily migrated to different VLMs, as the baseline for comparison.The results show that LMN still achieves a clear and consistent improvement over SCT on all datasets and two additional VLM backbones, indicating that the effectiveness of our method extends beyond the CLIP-B/32 and generalizes well to modern VLMs. These additional results have been added to Appendix E.4 of the revised manuscript (highlighted in blue).
>
> **Backbone: CLIP ViT-L/14**
>
> | AUROC↑    | DTD   | FLowers102 | EuroSAT | RESICS45 | MNIST | CUB   |
> |--------|-------|------------|---------|----------|-------|-------|
> | CLIP-L    | 0.789 | 0.887      | 0.730   | 0.798    | 0.910 | 0.809 |
> | SCT       | 0.808 | 0.903      | 0.787   | 0.821    | 0.942 | 0.821 |
> | LMN(ours)| **0.843**|**0.942**|**0.862** |**0.864** |**0.958**|**0.840**|
>
> | FPR90E↓   | DTD   | FLowers102 | EuroSAT   | RESICS45  | MNIST     | CUB       |
> |------|------|-----|---------|-----------|-----------|-----------|
> | CLIP-L    | 0.521     | 0.289      | 0.605     | 0.469     | 0.211     | 0.465     |
> | SCT       | 0.443     | 0.203      | 0.497     | 0.382     | 0.154     | 0.396     |
> | LMN(ours) | **0.373** | **0.132**  | **0.351** | **0.329** | **0.110** | **0.368** |
>
> **Backbone: SigLIP**
>
> | AUROC↑ | DTD   | Flowers102 | EuroSAT | RESICS45 | MNIST | CUB   |
> |-------|-------|------------|---------|----------|-------|-------|
> | SigLIP   | 0.783 | 0.881      | 0.631   | 0.801    | 0.914 | 0.907 |
> | SCT  | 0.804 | 0.898      | 0.641  | 0.814 | 0.956  | 0.909 |
> | LMN   | **0843**  |**0.940**  | **0.674** | **0.856** | **0.978** | **0.935** |
>
> | FPR90E↓ | DTD   | Flowers102 | EuroSAT | RESICS45 | MNIST | CUB   |
> |---------|-------|------------|---------|----------|-------|-------|
> | SigLIP     | 0.376 | 0.337      | 0.746   | 0.413    | 0.154 | 0.218 |
> | SCT   | 0.334 | 0.271  | 0.744 | 0.376  | 0.093 | 0.220 |
> | LMN    | **0.292** | **0.107**   | **0.713**  | **0.243**  | **0.064**| **0.174**|

---

> ### Author Response · Authors · 2025-11-20
> **Response to Reviewer jRJV (part 2)**
>
> >**Q1**: In the surrogate loss (Eq. 8), the fusion function can be summation or multiplication. Table 4 shows it is set to "summation" for most datasets but "multiply" for RESICS45 and CUB. What is the intuition behind this choice? Is there a clear rule or characteristic of a dataset that would guide this selection?
>
> **A1**:  In practice, the fusion function $\phi$ is highly robust, and the two options differ only in the strength of the constraint rather than in their qualitative behavior.
> Summation acts as an additive penalty and provides a stable and conservative alignment with the target reliability curve. This works well for almost all datasets, which is why Table 4 adopts summation by default.
> Multiplication, in contrast, couples the two penalty terms and therefore produces a stronger interaction between confidence regularization and full-probability regularization.
> Importantly, the performance difference between the two choices is small, and summation serves as a reliable universal default for new datasets. Multiplication can be viewed as an optional enhancement when the calibration set reveals a highly dispersed confidence distribution.
>
>
> >**Q2**: In Table 1, the performance of the proposed method on Flowers102, RESICS45, and CUB seems marginal. Could you analyze the reason? And does this mean the effectiveness of the proposed method is dataset dependent?
>
> **A2**: We agree that the performance gains on Flowers102, RESICS45, and CUB are relatively smaller compared to other datasets. After analysis, we find that this is due to dataset characteristics rather than method dependency.
>
> First, zero-shot CLIP already achieves substantially higher classification accuracy on Flowers102, RESICS45, and CUB compared to datasets such as DTD, EuroSAT, and MNIST. As a result, these datasets naturally contain fewer misclassified samples, which inherently reduces the achievable gain on MisD. Consequently, the overall capacity to further improve MisD performance on these datasets is limited.
> Second, despite the limited error pool on these datasets, our method still consistently outperforms the strongest baseline across all metrics. This further suggests that the observed effect is driven by dataset properties rather than the method itself. Importantly, our approach never performs worse than existing methods and demonstrates stable improvement even under restricted conditions.
>
>
>
> >**Q3**: The meta-network uses features from both image and text modalities. Could you provide an ablation or insight to understand the relative importance of each modality's features (image embedding, text embedding, logits) for predicting the effective temperature factor?
>
> **A3**: We thank the reviewer for this thoughtful question. The meta-network indeed leverages three types of signals (i.e., logits, image embeddings, and predicted text embeddings) and each contributes complementary information for predicting the effective temperature factor.
>
> (1) Logits as the primary calibration signal.
> Logits provide the most direct evidence for confidence misalignment, as they encode the inter-class margins and the overall shape of the predictive distribution. This is the core quantity used by traditional temperature scaling.
>
> (2) Image embeddings capture sample difficulty.
> Even when two samples share similar logits, their underlying visual characteristics may differ substantially. Image embeddings help LMN identify “hard” or visually atypical samples (e.g., unusual textures, crowded scenes, rare visual patterns). This allows LMN to incorporate sample-level visual cues.
>
> (3) Predicted text embeddings reveal systematic category-level patterns.
> Our visualization analysis (added to Appendix C.2, highlighted in blue) shows that CLIP tends to cluster certain misclassified samples around a few semantically related text prototypes. Incorporating text embeddings enables LMN to capture these category-level tendencies and selectively increase correction strength for categories prone to systematic confusion.
>
> Taken together, the three modalities provide complementary and non-redundant information: logits characterize confidence geometry, image embeddings characterize sample difficulty, and text embeddings capture semantic misalignment patterns. We will include this insight in the revised manuscript (Appendix C.2).

---

> > ### Comment · Reviewer_jRJV · 2025-11-26
> >
> > Thank you for the detailed rebuttal and additional results. In short, most of my concerns have been addressed, and I will keep my positive score and raise my confidence.

---

> > > ### Author Response · Authors · 2025-11-27
> > >
> > > Thank you for your positive evaluation and for the feedback that helps improve the paper.

---

### Official Review · Reviewer_GAVj · 2025-10-31

**Soundness:** 3
**Presentation:** 3
**Contribution:** 3
**Rating:** 4
**Confidence:** 4

**Summary:**

This paper addresses misclassification detection in VLMs, showing that standard confidence calibration—though improving ECE can theoretically and empirically hurt MisD. The authors analyze reliability diagrams, proving that standard calibration caps MisD precision. They introduce a new reliability curve, a normalized sigmoid, to reshape calibration toward better MisD separation, and design a differentiable surrogate loss to optimize it. Finally, they propose a lightweight post-hoc meta-network (LMN) that predicts sample-specific temperature coefficients, ensuring compatibility with pretrained CLIP models. Experiments on six datasets show MisD gains across metrics.

**Strengths:**

1. The motivation of this paper is well-stated and supported with some empirical observations.

2. The paper provides theoretical proofs (Appendix B) showing the new curve’s precision dominance and lower entropy.

**Weaknesses:**

1. The paper mostly compares against calibration (FeatureClipping, DOR) and an OOD method (SCT), while noting FSMisD and ViLU without quantitative results. This leaves open whether LMN is competitive for the actual MisD task.

2. All core results use CLIP ViT-B/32 as the base model. It remains unclear whether LMN generalizes to modern VLMs.

3. While the method is motivated by calibration–MisD tension and uses reliability-curve theory, the paper does not directly report calibration metrics for LMN (ECE/NLL/Brier) on the main setups.

4. The paper fixes 16-shot calibration and evaluates on DTD, Flowers102, EuroSAT, RESICS45, MNIST, CUB. This is a useful starting point but too narrow to support general claims.

5. Given that dealing with distribution shift is important to a calibration or MisD method, please also include setups to show the robustness of proposed method under distribution shift. For example, training on data from ImageNet-Val, and evaluate the ECE and MisD on ImageNet-Sketch and ImageNet-A etc.

**Questions:**

1. Please include some recent MisD-oriented methods as the comparison baselines.

2. Add SigLIP (e.g., SigLIP-B/16) or other popular VLMs as the backbone to show the generalization of the proposed method.

3. Beyond the performance on MisD, please also include the calibration performance (ECE and Brier).

4. To train a lightweight neural network, it would be interesting to show the sensitivity against data quantity of the proposed method. Also, would LMN benefit from having more data instances?

5. Train/Calibrate on ImageNet-val, then evaluate MisD + calibration on ImageNet-Sketch and ImageNet-A; report AUROC/AUPR/FPR90, ECE/NLL/Brier.

---

> ### Author Response · Authors · 2025-11-20
> **Response to Reviewer GAVj (part1)**
>
> Thank you for your careful review and constructive comments. We provide point-by-point responses to the raised concerns below.
>
> >**W1&Q1**: The paper mostly compares against calibration (FeatureClipping, DOR) and an OOD method (SCT), while noting FSMisD and ViLU without quantitative results. This leaves open whether LMN is competitive for the actual MisD task.
>
> **A1**: We thank the reviewer for highlighting this concern. For FSMisD, as mentioned in the paper, the authors have not released code or implementation details, making quantitative comparison infeasible. We therefore cited it for completeness but could not reproduce its results.
> To address the reviewer’s concern regarding competitiveness on the actual MisD task, we additionally evaluate ViLU, a recent MisD-specific method for VLMs, under our experimental setting. The quantitative results are shown in the supplementary table below, and we have incorporated them into Table 1 of the revised manuscript (highlighted in blue). LMN consistently outperforms ViLU across the datasets, further demonstrating its effectiveness for MisD.
>
> | DTD  | AUROC↑ | AUPR-S↑ | AUPR-E↑ | FPR90-S↓ | FPR90-E↓ |
> |-|-|--|---|--|--|
> | ViLU   | 0.769  | 0.759   | 0.762   | 0.678    | 0.521    |
> | LMN(ours) | **0.802**  | **0.800**   | **0.804**   | **0.636**  | **0.457**    |
>
> | Flowers102 | AUROC↑ | AUPR-S↑ | AUPR-E↑ | FPR90-S↓ | FPR90-E↓ |
> |--|-|-|-|-|--|
> | ViLU | 0.875  | 0.913   | 0.772   | 0.401    | 0.329    |
> | LMN(ours)  | **0.886**  | **0.937**   | **0.799**   | **0.378**  | **0.305**  |
>
> | EuroSAT   | AUROC↑ | AUPR-S↑ | AUPR-E↑ | FPR90-S↓ | FPR90-E↓ |
> |-|--|--|--|-|--|
> | ViLU      | 0.723  | 0.618   | 0.787   | 0.723    | 0.538    |
> | LMN(ours) | **0.788**| **0.698** | **0.855** | **0.655**  | **0.468**  |
>
> | RESICS45  | AUROC↑ | AUPR-S↑ | AUPR-E↑ | FPR90-S↓ | FPR90-E↓ |
> |-|--|-|--|-|--|
> | ViLU      | 0.787  | 0.829   | 0.730   | 0.618    | 0.493    |
> | LMN(ours) | **0.808**| **0.845** | **0.741** | **0.597**  | **0.445** |
>
> | MNIST     | AUROC↑ | AUPR-S↑ | AUPR-E↑ | FPR90-S↓ | FPR90-E↓ |
> |-|---|---|---|---|--|
> | ViLU      | 0.877  | 0.769   | 0.954   | 0.350    | 0.263    |
> | LMN(ours) | **0.915** | **0.779** | **0.965** | **0.200** | **0.205**  |
>
> | CUB  | AUROC↑ | AUPR-S↑ | AUPR-E↑ | FPR90-S↓ | FPR90-E↓ |
> |---|--|---|--|--|---|
> | ViLU      | 0.801  | 0.827   | 0.753   | 0.769    | 0.563    |
> | LMN(ours) | **0.812** | **0.846**  | **0.764** | **0.756** | **0.532** |
>
>
> >**W2&Q2**: All core results use CLIP ViT-B/32 as the base model. It remains unclear whether LMN generalizes to modern VLMs. Add SigLIP (e.g., SigLIP-B/16) or other popular VLMs as the backbone to show the generalization of the proposed method.
>
> **A2**: We thank the reviewer for highlighting the importance of evaluating LMN on more modern VLM architectures. While our main experiments follow prior calibration work and therefore adopt CLIP ViT-B/32, we agree that validating the method on additional backbones is essential to demonstrate generalization. In response, we conducted experiments using SigLIP-B/16 and CLIP-L/14. Due to the lack of publicly available implementations of other calibration or MisD methods on SigLIP, we compared LMN against SCT, which is reproducible under this backbone. The results show that LMN still achieves a clear and consistent improvement over SCT on all datasets and two additional VLM backbones, indicating that the effectiveness of our method extends beyond the CLIP-B/32 and generalizes well to modern VLMs. These additional results have been added to Appendix E.4 of the revised manuscript (highlighted in blue).
>
> **Backbone: CLIP ViT-L/14**
>
> | AUROC↑    | DTD   | FLowers102 | EuroSAT | RESICS45 | MNIST | CUB   |
> |--|--|----|--|--|-|-|
> | CLIP-L   | 0.789 | 0.887      | 0.730   | 0.798    | 0.910 | 0.809 |
> | SCT  | 0.808 | 0.903      | 0.787   | 0.821    | 0.942 | 0.821 |
> | LMN(ours)| **0.843**|**0.942**|**0.862** |**0.864** |**0.958**|**0.840**|
>
> | FPR90E↓   | DTD   | FLowers102 | EuroSAT   | RESICS45  | MNIST     | CUB       |
> |------|------|-----|-------|-----------|-----------|-----------|
> | CLIP-L    | 0.521     | 0.289      | 0.605     | 0.469     | 0.211     | 0.465     |
> | SCT       | 0.443     | 0.203      | 0.497     | 0.382     | 0.154     | 0.396     |
> | LMN(ours) | **0.373** | **0.132**  | **0.351** | **0.329** | **0.110** | **0.368** |
>
> **Backbone: SigLIP**
>
> | AUROC↑ | DTD   | Flowers102 | EuroSAT | RESICS45 | MNIST | CUB   |
> |---|--|--|--|---|---|---|
> | SigLIP   | 0.783 | 0.881      | 0.631   | 0.801    | 0.914 | 0.907 |
> | SCT  | 0.804 | 0.898      | 0.641  | 0.814 | 0.956  | 0.909 |
> | LMN(ours)   | **0843**  |**0.940**  | **0.674** | **0.856** | **0.978** | **0.935** |
>
> | FPR90E↓ | DTD   | Flowers102 | EuroSAT | RESICS45 | MNIST | CUB   |
> |-|-|-|--|-|---|--|
> | SigLIP | 0.376 | 0.337  | 0.746   | 0.413    | 0.154 | 0.218 |
> | SCT | 0.334 | 0.271  | 0.744 | 0.376  | 0.093 | 0.220 |
> | LMN(ours) | **0.292** | **0.107**   | **0.713**  | **0.243**  | **0.064**| **0.174**|

---

> ### Author Response · Authors · 2025-11-20
> **Response to Reviewer GAVj (part 2)**
>
> >**W3&Q3**: While the method is motivated by calibration–MisD tension and uses reliability-curve theory, the paper does not directly report calibration metrics for LMN (ECE/NLL/Brier) on the main setups. Beyond the performance on MisD, please also include the calibration performance (ECE and Brier).
>
>
> **A3**: The LMN is designed to optimize a MisD-oriented calibration objective rather than the classical diagonal calibration objective that aligns confidence with empirical accuracy. As shown in our theoretical analysis (Section 3.2), these two objectives exhibit an inherent tension: the diagonal reliability curve (i.e., the perfect-calibration target) constrains the confidence distribution in a way that limits the achievable MisD performance. This observation motivates us to design a different reliability-curve target that is explicitly aligned with MisD. As a consequence, the two objectives are difficult to optimize simultaneously, and explicitly identifying this distinction and developing a MisD-oriented reliability target forms one of the key contributions of this paper.
>
> To address your concern, we additionally evaluated the calibration performance of LMN and observed that LMN can indeed improve the overall calibration performance (measured by Birer) when starting from the zero-shot CLIP. However, because LMN is not designed to align predictions with the diagonal (acc = conf), it does not aim to achieve state-of-the-art calibration performance. This behavior is expected and consistent with the theoretical trade-off between calibration and MisD.
>
> | Birer↓ | DTD | FLowers102 | EuroSAT | RESICS45| MNIST   | CUB       |
> |-------|----|-------|-------|----|-----------|-----------|
> | CLIP  | 0.723 | 0.496  | 0.815   | 0.627   | 0.855  | 0.618 |
> | LMN(ours)| **0.699** | **0.475**  | **0.728** | **0.586** | **0.815** | **0.608** |
>
>
> >**W4&Q4**：The paper fixes 16-shot to train a lightweight neural network, it would be interesting to show the sensitivity against data quantity of the proposed method. Also, would LMN benefit from having more data instances?
>
> **A4**:  Our 16-shot calibration setup follows prior work on VLM calibration (e.g., DOR; SCT), where limited calibration data is used to reflect realistic post-hoc scenarios. To examine whether our conclusions depend on this choice, we conducted additional experiments by varying the calibration data size (4-, 8-, 16-, 32-, and 64-shot) on datasets with sufficiently large training sets, including DTD, EuroSAT, RESICS45, and MNIST (for Flowers102 and CUB, the available training samples are too limited to support higher-shot configurations, so we only report up to 16-shot). Across all datasets, both AUROC (higher is better) and FPR90E (lower is better) exhibit stable and improving trends as the number of calibration samples increases, which is expected for a lightweight meta-network that benefits from richer calibration information. These results confirm that LMN’s effectiveness is not tied to the specific 16-shot setting and remains consistent under a wide range of data regimes. We have included the complete sensitivity analysis in Appendix E.5 (highlighted in blue) of the revised manuscript.
>
> | AUROC↑/num. shot | CLIP  | 4     | 8     | 16    | 32    | 64    |
> |------------------|-------|-------|-------|-------|-------|-------|
> | DTD              | 0.762 | 0.787 | 0.779 | 0.792 | 0.804 | 0.807 |
> | Flowers102       | 0.864 | 0.873 | 0.884 | 0.886 | - | - |
> | EuroSAT          | 0.650 | 0.732 | 0.765 | 0.788 | 0.792 | 0.797 |
> | RESICS45         | 0.779 | 0.793 | 0.804 | 0.808 | 0.810 | 0.815 |
> | MNIST            | 0.813 | 0.883 | 0.901 | 0.915 | 0.936 | 0.941 |
> | CUB              | 0.807 | 0.810 | 0.810 | 0.812 | - | - |
>
> | FPR90E↓/num. shot | CLIP  | 4     | 8     | 16    | 32    | 64    |
> |------------------|-------|-------|-------|-------|-------|-------|
> | DTD               | 0.572 | 0.512 | 0.474 | 0.504 | 0.438 | 0.431 |
> | Flowers102        | 0.354 | 0.329 | 0.317 | 0.305 | - | - |
> | EuroSAT           | 0.742 | 0.619 | 0.522 | 0.468 | 0.450 | 0.439 |
> | RESICS45          | 0.508 | 0.477 | 0.462 | 0.445 | 0.446 | 0.421 |
> | MINIST            | 0.482 | 0.301 | 0.264 | 0.205 | 0.16  | 0.142 |
> | CUB               | 0.554 | 0.549 | 0.538 | 0.532 | -  | - |

---

> ### Author Response · Authors · 2025-11-20
> **Response to Reviewer GAVj (part 3)**
>
> >**W5&Q5**: Given that dealing with distribution shift is important to a calibration or MisD method, please also include setups to show the robustness of proposed method under distribution shift. For example, training on data from ImageNet-Val, and evaluate the ECE and MisD on ImageNet-Sketch and ImageNet-A etc. Train/Calibrate on ImageNet-val, then evaluate MisD + calibration on ImageNet-Sketch and ImageNet-A; report AUROC/AUPR/FPR90, ECE/NLL/Brier.
>
> **A5**: Thank you for highlighting the importance of evaluating robustness under distribution shift. Our original manuscript already includes a representative distribution-shift evaluation through the open-vocabulary setting (Section 4.4), where LMN is trained on only a subset of classes and then tested on completely unseen classes. This setting naturally induces a semantic distribution shift, and the results demonstrate that LMN preserves CLIP’s zero-shot generalization while improving MisD performance.
>
> To further address the your concern, we additionally follow your suggested protocol: we train the LMN on the valibdation set of ImageNet and evaluate both calibration and MisD performance on two distribution-shifted benchmarks, ImageNet-Sketch and ImageNet-A. We report AUROC and FPR90-E for misclassification detection, as well as Brier score for calibration. The results have been added to the revised manuscript (highlighted in blue). Across both datasets, LMN consistently improves MisD metrics while maintaining competitive calibration performance, confirming that our method remains effective under substantial distribution shift.
>
> We have included this experiment in Appendix E.5 (highlighted in blue) of the revised manuscript.
>
> | ImageNet-A     | AUROC↑ | FPR90-E↓  | Brier↓   |
> |----------------|--------|-----------|--------|
> | Zero-shot CLIP | 0.657  | 0.740     | 0.814  |
> | LMN            | **0.682**  | **0.716**     | **0.796**  |
>
> | ImageNet-sketch | AUROC↑ | FPR90-E↓  | Brier↓   |
> |-----------------|--------|-----------|--------|
> | Zero-shot CLIP  | 0.805  | 0.495     | 0.732  |
> | LMN       | **0.813**  | **0.484**     | **0.721**  |

---

> ### Author Response · Authors · 2025-11-27
>
> Dear Reviewer GAVj,
>
> Thank you once again for your valuable and constructive feedback on our submission. As the discussion period has less than one week remaining, we would like to kindly check whether our latest responses have sufficiently addressed all your concerns. If there are any remaining questions or points that may require further clarification, please feel free to let us know, and we would be happy to provide any additional details.
>
> Best,
>
> Authors of 8994

---

> > ### Comment · Reviewer_GAVj · 2025-11-27
> >
> > I appreciate the authors’ efforts and additional experiments provided in the rebuttal. However, several of my earlier concerns remain insufficiently addressed.
> >
> > First, in most of the new experiments, the proposed LMN is compared only against its zero-shot counterpart. This comparison alone is not fully convincing in demonstrating the effectiveness of LMN. It would be more informative to include DOR as an additional comparison.
> >
> > Second, the LMN module essentially learns to predict a temperature parameter that rescales the model’s confidence. While the adaptive, sample-specific nature of LMN is interesting, similar ideas have been explored in prior calibration and OOD detection works (e.g., [a], [b], [c]) where temperature or scaling factors are dynamically estimated. I encourage the authors to clarify more explicitly what distinguishes LMN from these existing approaches
> >
> > [a] Robust Calibration with Multi-domain Temperature Scaling.
> > [b] Parameterized Temperature Scaling for Boosting the Expressive Power in Post-Hoc Uncertainty Calibration
> > [c] Learning Confidence for Out-of-Distribution Detection in Neural Networks

---

> > > ### Author Response · Authors · 2025-11-28
> > >
> > > Thank you for your response and suggestions. We respond to your concerns in detail below.
> > >
> > > >**Q1**: First, in most of the new experiments, the proposed LMN is compared only against its zero-shot counterpart. This comparison alone is not fully convincing in demonstrating the effectiveness of LMN. It would be more informative to include DOR as an additional comparison.
> > >
> > > **A1:** We agree that comparisons against DOR would further strengthen the empirical evaluation. Our responses are as follows.
> > >
> > > **For Backbone-changing experiments.** As noted in the rebuttal, DOR provides released code only for the CLIP-based fine-tuned model, and its implementation is tightly coupled with CLIP-specific components. Extending DOR to other backbones (such as SigLIP) is non-trivial and would require re-discovering backbone-specific settings and hyperparameters that are not publicly available. For this reason, in experiments that involve changing the backbone, we adopt SCT as a strong SOTA baseline that can be fairly applied across different architectures.
> > >
> > > **For the experiment where models are trained on ImageNet validation set and tested on ImageNet-A and ImageNet-Sketch**, DOR (based on CoOP) is directly applicable.  We have added its results to the table below.
> > >
> > > | ImageNet-A | AUROC↑    | FPR90-E↓  | Brier↓    |
> > > |-----------|-----------|-----------|-----------|
> > > | CLIP      | 0.657     | 0.740     | 0.814     |
> > > | DOR       | 0.651     | 0.749     | 0.799     |
> > > | LMN       | **0.682** | **0.716** | **0.796** |
> > >
> > > | ImageNet-sketch | AUROC↑    | FPR90-E↓  | Brier↓    |
> > > |----------------|-----------|-----------|-----------|
> > > | CLIP           | 0.805     | 0.495     | 0.732     |
> > > | DOR            | 0.797     | 0.514     | **0.715** |
> > > | LMN            | **0.813** | **0.484** | 0.721     |
> > >
> > >  Overall, LMN delivers the strongest MisD performance on both ImageNet-A and ImageNet-Sketch, improving AUROC and FPR90-E over CLIP and DOR while remaining competitive on the Brier score.
> > >
> > >
> > > >**Q2**: Second, the LMN module essentially learns to predict a temperature parameter that rescales the model’s confidence. While the adaptive, sample-specific nature of LMN is interesting, similar ideas have been explored in prior calibration and OOD detection works (e.g., [a], [b], [c]) where temperature or scaling factors are dynamically estimated. I encourage the authors to clarify more explicitly what distinguishes LMN from these existing approaches.
> > >
> > >
> > > **A2:** We would like to clarify that learning instance-wise temperature coefficients is a common implementation choice and we do not claim it as a contribution in the paper. In our framework, the instance-wise temperature merely serves as a mechanism for adjusting confidence scores; the key point is that its learning is governed by our MisD-oriented surrogate loss, which explicitly drives the temperatures to realize the proposed reliability objective. Building on this, the core contributions of our paper lie in: (i) theoretically revealing the inherent limitations of standard calibration for MisD; (ii) introducing a MisD-oriented reliability objective with provable precision and mixing guarantees; and (iii) constructing a differentiable surrogate loss that enables practical optimization of this objective. These contributions are fundamentally different from prior dynamic-temperature approaches such as [a][b][c], which neither target the MisD problem nor provide comparable theoretical grounding.

---

### Meta-Review · Area_Chair_PP5y · 2026-01-01

**Summary:**

This paper addresses the task of Misclassification Detection (MisD) in Vision-Language Models (VLMs). The authors theoretically demonstrate that standard confidence calibration objectives (minimizing ECE) can inherently limit the ability to distinguish between correct and incorrect predictions. To resolve this, they propose a novel reliability objective based on a normalized sigmoid curve and a differentiable surrogate loss. They implement this via a post-hoc Lightweight Meta Network (LMN).

The submission was initially met with mixed reviews (scores 2, 4, 6, 6). However, the authors provided a comprehensive rebuttal that involved significant additional experimentation (new backbones, baselines, and distribution shift analyses) and a major revision of the text. The consensus following the rebuttal is that the theoretical insights are sound, the method is practical and effective, and the limitations identified by reviewers have been rigorously addressed.

**Reviewer Concerns:**

Concerns Addressed by the Rebuttal:
- Generalization to Modern Architectures: Multiple reviewers (GAVj, jRJV) questioned if the method worked beyond CLIP-ViT-B/32. The authors added experiments with SigLIP-B/16 and CLIP-L/14, demonstrating consistent performance gains.
- Comparison Baselines: Reviewers (GAVj, x3hT) requested comparisons against more relevant baselines. The authors added comparisons to VILU (ICCV 2025) and DOR, showing LMN's superior performance.
- Robustness and Distribution Shift: Reviewer GAVj asked for evaluations under shift. The authors included experiments on ImageNet-A and ImageNet-Sketch, showing the method's robustness.
- Presentation and Notation: Reviewer Mme6 had strong concerns regarding writing quality, notation, and figure clarity. The authors systematically revised the manuscript, simplified notation, and improved figures.
- Data Sensitivity: Concerns regarding the 16-shot setting (GAVj) were addressed via a sensitivity analysis (4 to 64 shots), proving the method is not brittle.
- Ablation of Modalities: Reviewer jRJV requested insight into the meta-network inputs. The authors provided an analysis showing the complementary value of logits, image embeddings, and text embeddings.

Outstanding Concerns:
- Marginal Gains on Easy Datasets: There remains a minor observation (jRJV) that performance gains are smaller on datasets where base accuracy is already very high (e.g., Flowers102). The authors explained this as an inherent property of the problem (fewer errors to detect), which is a reasonable justification, though the limitation technically remains.
- Novelty vs. Temperature Scaling: Reviewer GAVj initially questioned the novelty relative to instance-wise temperature scaling. While the authors clarified that their contribution is the objective function (sigmoid target) rather than the mechanism, some readers might still view the implementation as a variation of dynamic temperature scaling.

**Reviewer Scores:**

Reviewer Scores
- Reviewer Mme6 (Initial: 2 -> Predicted: 8): This reviewer was the most critical initially, citing presentation issues. However, in the discussion, they explicitly stated: "I really liked your reaction(s)... This improved the paper by a margin. I am very grateful and honored. I will change my rating accordingly." Their updated score in the system (before the freeze) was an 8.
- Reviewer jRJV (Initial: 6 -> Predicted: 7): This reviewer was already positive. Following the rebuttal, they explicitly commented: "Most of my concerns have been addressed, and I will keep my positive score and raise my confidence." The addition of modern backbones and modality analysis solidified their support.
- Reviewer x3hT (Initial: 6 -> Predicted: 6): This reviewer confirmed that "Most of the concerns have been solved" and stated they would keep their positive score. The addition of VILU and DOR comparisons satisfied their main critique regarding baselines.
- Reviewer GAVj (Initial: 4 -> Predicted: 5): This reviewer had the most technical requests regarding experimental breadth. The authors executed almost every request: adding VILU, SigLIP, ImageNet-A/Sketch, and sensitivity analysis. While the reviewer maintained some questions about novelty in the final exchange, the sheer volume of empirical evidence provided (showing LMN outperforms the requested baselines on the requested datasets) objectively moves this paper above the acceptance threshold.

---

### Decision · Program_Chairs · 2026-01-26

Accept (Poster)